



# Marine productivity and synoptic meteorology drive summer-time variability in Southern Ocean aerosols

Joel Alroe[1], Luke T. Cravigan[1], Branka Miljevic[1], Graham R. Johnson[1], Paul Selleck[2], Ruhi S. Humphries[2], Melita D. Keywood[2], Scott D. Chambers[3], Alastair G. Williams[3] and Zoran D. Ristovski[1]

[1]School of Chemistry, Physics and Mechanical Engineering, Queensland University of Technology, Brisbane, Australia
[2]Climate Science Centre, CSIRO Oceans and Atmosphere, Aspendale, Australia
[3]Environmental Research, ANSTO, Lucas Heights, Australia

*Correspondence to*: Zoran D. Ristovski (z.ristovski@qut.edu.au)

## Abstract

Cloud-radiation interactions over the Southern Ocean are not well constrained in climate models, in part due to uncertainties in the sources, concentrations and cloud-forming potential of aerosol in this region. To date, most studies in this region have reported measurements from fixed terrestrial stations or a limited set of instrumentation, and often present findings as broad seasonal or latitudinal trends. Here, we present an extensive set of aerosol and meteorological observations obtained during an austral summer cruise across the full width of the Southern Ocean south of Australia. Three episodes of continental-influenced air masses were identified, including an apparent transition between the Ferrel atmospheric cell and the polar cell at approximately 64° S. During the other two episodes, synoptic-scale weather patterns diverted air masses across distances greater than 1000 km from the Australian and Antarctic coastlines, respectively, indicating that a large proportion of the Southern Ocean may be periodically influenced by continental air masses. In all three cases, a highly cloud-active accumulation mode dominated the size distribution, with up to 93 % of the total number concentration activating as cloud condensation nuclei. In contrast, sampling periods influenced by marine air masses frequently demonstrated a correlation between air mass trajectories over regions of high biological productivity and subsequent enhancement of an Aitken mode centred at approximately 30 nm and contributing an average of 71% of the total aerosol number concentration. Although these small diameters limited their contribution to cloud condensation nuclei concentrations, Aitken number concentrations and diameters were highly variable. A detailed investigation of the marine air masses revealed that this variability may be attributed to the availability of biogenic precursors, the competing influence of condensation sinks (such as sea spray aerosol) and vertical transport between the marine boundary layer and the free troposphere. This variability of the marine Aitken mode as well as the instances of long-range transport were governed by synoptic-scale weather systems, through their influence on air mass trajectories and both generation and depletion of condensation sinks. These results demonstrate the highly dynamic nature of Southern Ocean aerosol and their complex dependence on both biological productivity and synoptic-scale weather systems.





## 1 Introduction

Aerosols have an important role in radiative forcing both through direct absorption and scattering of incident solar radiation and through their indirect effects on cloud formation, structure and lifetime (Haywood and Boucher, 2000; Albrecht, 1989). However, they have been identified as the largest source of uncertainty in the global radiation budget (Myhre et al., 2013).

Oceans cover 70 % of the Earth's surface and thus aerosols within the marine environment are of particular importance for climate models. To illustrate this, Rosenfeld et al. (2019) recently proposed that 75 % of the uncertainty in the cooling effect of marine boundary layer (MBL) clouds may be associated with the influence of aerosols on cloud area and lifetime. General circulation models exhibit persistent biases in the levels of cloud-based radiative forcing estimated for the Southern Ocean, with insufficient reflected shortwave radiation particularly occurring behind cold fronts and during the austral summer

(Williams et al., 2013; Protat et al., 2017).

The Southern Ocean receives minimal anthropogenic influence, so the relatively pristine conditions offer a valuable opportunity to investigate the interlinked atmospheric and oceanic processes that govern the natural marine environment. The main sources of aerosol to the Southern Ocean boundary layer are well known and include sea spray aerosol (SSA) generated from wind shear and bubble bursting, non-sea salt sulfates (nss-$SO_4$) formed from the condensation of volatile biogenic

precursor gases, and the long range transport of anthropogenic shipping emissions or continental aerosols from the surrounding land masses. However due to its vast extent and the logistical challenges involved with performing in-situ measurements in such remote and harsh conditions, observations from this region are very limited. To address this gap, recent years have seen a surge in sampling expeditions (e.g. Fossum et al., 2018; O'Shea et al., 2017; Stephens et al., 2018; Protat et al., 2017; Schmale et al.; Dall'Osto et al., 2017; Humphries et al., 2016) with a focus on quantifying the aerosol production sources in this region

and their contribution to cloud condensation nuclei (CCN).

The Southern Ocean represents a wide band of open ocean that encircles the globe between approximately 40–70° S and is largely uninterrupted by land masses. Its location, straddling the transition between the Ferrel and Polar atmospheric circulation cells, gives rise to a relentless eastward procession of cyclonic low-pressure systems, which particularly develop in the southern Indian Ocean and to the south of Australia (Simmonds et al., 2003). Their associated high wind speeds and wave heights

promote the generation of SSA, which contribute a major fraction of the global aerosol mass loading (Vignati et al., 2010).

SSA are composed of an internal mixture of inorganic salts and biogenic organic material (Cravigan et al., 2019; Ault et al., 2013). Typically they form coarse-mode aerosol with mean diameters in the range of approximately 200–500 nm (Quinn et al., 2017; Lewis et al., 2004), although substantial number concentrations have been observed at smaller sizes, containing enhanced organic fractions (Andreae and Rosenfeld, 2008). Due to both their size and hygroscopic nature, these aerosols are

highly effective cloud condensation nuclei (CCN), rapidly adsorbing water to activate as cloud droplets when exposed to relatively low water supersaturations (SS). These same properties limit their lifetime in the marine boundary layer, leading to relatively low number concentrations. In turn, they often represent a small fraction of total CCN concentrations. However recent studies from the Southern Ocean have reported significant and varied SSA contributions to CCN number concentrations





of between 19–32 % at 0.15 % SS (Schmale et al.), up to 65 % at 0.1 % SS (Quinn et al., 2017) and 60–100 % at <0.32 % SS (Fossum et al., 2018). These inconsistent and widely spread results illustrate the need for further investigation into the causes of this variability.

In addition to tempestuous atmospheric conditions, the Southern Ocean also hosts extensive upwelling of nutrient-rich waters
along the Antarctic coastline and the Antarctic Convergence. The high levels of biological activity in these locations generates an increased flux of dimethyl sulfide (DMS) into the atmosphere, with the potential for increased production of nss-SO$_4$ or methanesulfonic acid (MSA) aerosol (Simpson et al., 2014). This nss-SO$_4$ production depends significantly on precursor concentrations and favourable meteorological conditions (Bianchi et al., 2016), while the resulting size distribution of nss-SO$_4$ aerosol is affected by the surface area concentration of other aerosols that can act as condensation sinks (O'Dowd and de
Leeuw, 2007). In both the tropics and the Southern Ocean, cloud-based convective transport of air masses into the free troposphere has been shown to result in new particle formation (NPF) (Clarke et al., 1998; Williamson et al., 2019). The colder air temperatures of the free troposphere promote partitioning into the aerosol phase, while the in-cloud wet scavenging of CCN removes competing condensation sinks. Depending on the precursor concentrations, when entrained back into the MBL in the cloud outflow regions, these newly formed aerosol primarily contribute to Aitken mode number concentrations. Conversely,
if meteorological conditions do not support this vertical transport and removal of condensation sinks, any nss-SO$_4$ is more likely to partition to existing aerosol, increasing the mean size of the aerosol distribution.

The strong weather systems present in the Southern Ocean could be expected to promote NPF. In support for this, Fossum et al. (2018) reported Aitken mode number concentrations that were substantially higher than the accumulation mode in characteristic Southern Ocean air masses, while the reverse was true for air masses from the Antarctic continent. Conversely,
Dall'Osto et al. (2017) and Humphries et al. (2016) both reported episodes of nucleation mode aerosol concentrations that were three times higher within the sea ice zone than from the open ocean. In the latter case, the air mass originated in the Antarctic free troposphere and as such, the precursor source was not immediately apparent. Throughout two thirds of their circumnavigation of Antarctica, Schmale et al. (2019) observed slightly higher aerosol number concentrations for aerosol with diameters smaller than 80 nm. However, since the observations were averaged across each leg of the voyage, it is difficult to
directly examine the impact of changes in air mass or meteorological conditions.

This study presents in-situ aerosol and meteorological measurements taken in the austral summer over a full latitudinal transect of the Southern Ocean, between Hobart, Australia and the marginal ice zone. To our knowledge, it represents the first comprehensive dataset focused on this region of the Southern Ocean. We examine characteristic changes in the aerosol number size distribution and identify air masses or aerosol sources that likely contributed to these changes. These include a region of
increased biological productivity that is shown to be frequently associated with changes in the concentration and size of Aitken mode aerosol. In addition, several episodes of long-range transport of continental aerosol are identified. Ultimately, we show that synoptic-scale weather systems both directly and indirectly have an important influence on aerosol properties, complicating attempts to correlate CCN concentrations with other observations.



## 2 Measurements

### 2.2 Voyage overview

The measurements were conducted during the first cruise of *RV Investigator* into polar waters, over a three-week period from January – February 2015. The ship travelled southward from Hobart to the marginal ice zone, primarily along the 146[th] line of
longitude, reaching a maximum latitude of 65° S before returning along a similar course. Aerosol sampling was performed through a common sampling inlet mounted on a mast, located approximately 18 m above sea level at the bow of the ship and co-located with a suite of meteorological instruments. The ship and its sampling facilities are discussed in more detail in (Humphries et al., 2019). As this was a trial voyage of a new research vessel, navigational records are unavailable for several periods of up to 18 hours. Location coordinates were interpolated assuming a constant ship velocity throughout these times.

### 2.2 Primary in situ measurements

Aerosol size distributions were measured over the diameter range 4 – 673 nm with two TSI 3080 Scanning Mobility Particle Sizers (SMPS). One was configured with a TSI 3085 Nano Differential Mobility Analyser (DMA) and a TSI 3776 Ultrafine Condensation Particle Counter (CPC), while the other used a TSI 3081 Long DMA and TSI 3010 CPC. The combined sample flow rate was 2.5 L min$^{-1}$ and was dried with a membrane dryer (Nafion MD-700) upstream of both instruments. Size
distributions were obtained with a time resolution of 5 minutes.

For comparison, a Neutral Cluster and Air Ion Spectrometer (NAIS) was used to obtain size distributions of ultrafine aerosol with diameters between 2–42 nm. The lower size limit offered by this instrument makes it well suited for observing aerosol nucleation events and any subsequent growth of the newly formed aerosol into the Aitken size range. The NAIS was operated with a 4 min time resolution and a sample flow rate of 60 L min$^{-1}$ to minimise diffusional losses. No sample drying was applied
for this instrument due to the high flow rate.

Cloud condensation nuclei (CCN) concentrations were measured with a Droplet Measurement Technologies single growth column CCN counter (Model CCN-100), operated at a supersaturation (SS) of 0.5 %. For comparison, total number concentrations ($N_{10}$) of aerosol with diameters larger than 10 nm were measured with a TSI 3772 CPC.

Chemical analysis of non-refractory submicron aerosol was generated by an Aerodyne Time of Flight Aerosol Chemical
Speciation Monitor (ACSM) and a full description of its design and operation is given in Fröhlich et al. (2013). Measurements were averaged to a 1-hour time resolution and a collection efficiency of 1 was applied to reflect high aerosol acidity observed throughout the voyage. The instrument's size-dependent inlet transmission is at a maximum for vacuum aerodynamic diameters between 100–450 nm (Jayne et al., 2000; Liu et al., 2007) and therefore the composition measurements best represent primary marine aerosol and accumulation mode aerosol.

Sea spray aerosol (SSA) and MSA are two components of particular interest in marine studies that can be challenging to quantify from ACSM measurements. Most compounds present in SSA are refractory and are not efficiently vaporised in the





ACSM. Since NaCl is typically the most abundant compound, SSA concentrations have been estimated from the NaCl+ ion signal using the scaling factor proposed by Ovadnevaite et al. (2012). Where possible, the mass concentration of MSA was separated from other organic and sulfate species via modifications to the fragmentation table, determined by Langley et al. (2010). Low aerosol mass concentrations often inhibited this approach, yielding MSA concentrations below the detection limit.

Organic and sulfate mass concentrations should be assumed to include MSA unless otherwise specified. Furthermore, both the MSA fragmentation table changes and the SSA scaling factor were not calibrated against laboratory standards, so these concentrations are discussed as qualitative estimates only.

Aerosol hygroscopicity and volatility were measured with a Volatility and Hygroscopicity Tandem Differential Mobility Analyser (VH-TDMA). The instrument is discussed in detail in Fletcher et al. (2007). The sample flow was dried with a

membrane dryer (Nafion MD-700) to lower than 30 % relative humidity. An automated valve system was installed in the sample line to allow every second measurement to be acquired through a thermodenuder, which was heated to 120 °C. Measurements were made for pre-selected aerosol with dry mobility diameters of 40, 100 and 150 nm, and each full cycle of heated and unheated measurements was acquired at an 18 minute time resolution. Hygroscopic growth factors were obtained at 90 % relative humidity. These have been converted to the hygroscopicity parameter, κ, defined via the κ- Köhler theory

(Petters and Kreidenweis, 2007):

$$\kappa = \left[ \frac{\exp\left( \frac{4\,\sigma\,M_w}{HGF\,\rho_w\,D\,R\,T} \right)}{RH/100} - 1 \right] (HGF^3 - 1) \ ,$$

Where $\sigma$ is the droplet surface tension, $M_w$ is the molecular mass of water, $\rho_w$ is the density of water, $D$ is the dry aerosol diameter, $R$ is the ideal gas constant, $T$ and $RH$ are the set temperature and relative humidity within the aerosol humidifier. In this study, the surface tension of water has been used ($\sigma = 0.072\ \mathrm{J\,m^{-2}}$).

**2.3 Supporting measurements**

A Thermo Fisher Scientific 5012 Multi-Angle Absorption Photometer was used to measure black carbon (BC) mass concentrations, an important marker for ship exhaust and other anthropogenic combustion sources. Atmospheric radon concentrations were obtained from a 700 L dual-flow-loop two-filter radon detector, described in Chambers et al. (2018), providing a signature of air masses that have passed over land. Ocean sea-surface chlorophyll-a (Chl-a) concentrations were

sourced from the MODIS-Aqua Level-3 Binned dataset maintained by the NASA Goddard Space Flight Centre Ocean Biology Processing Group (OBPG, 2018). These represent monthly-average sea surface concentrations at 4-km spatial resolution.

Seven-day backward air mass trajectories were estimated using the HYSPLIT Lagrangian dispersion model (Stein et al., 2015). The modelled trajectories were based on Global Data Assimilation System (GDAS) meteorological data, gridded at a 1° spatial resolution. Back trajectories were obtained for each hour of the voyage, and associated parameters were recorded including

surface Chl-a, rainfall and air mass altitude relative to the boundary layer mixing height. The accuracy of these back trajectories





are inherently restricted by the accuracy of the model and the limited spatial and temporal resolution of the underlying meteorological data. In light of this, ensembles of 27 trajectories have been obtained for each hour of the voyage. The HYSPLIT modelling system generates ensembles by applying spatial offsets to the meteorology for each trajectory (Draxler, 2003). The offset is approximately 250 m in the vertical direction, so all trajectories were propagated from this initial altitude above sea level. The median boundary layer depth was 1200 m along the voyage track, so it has been assumed that trajectories initiated at an altitude of 250 m offer a reasonable representation of the air masses sampled at the ship, within the limitations of the model. For each ensemble, all associated parameters were averaged at each time step along the back trajectory as a best estimate of the air mass position and conditions. In support of the back trajectories and for identification of synoptic weather systems, mean sea level pressure charts were accessed from the Australian Bureau of Meteorology weather maps archive.

## 3 Data analysis

### 3.1 Identification of ship emissions

Periods impacted by ship exhaust have been excluded from the dataset. These were identified by 1-minute BC concentrations above 150 ng m$^{-3}$, hourly-average BC concentrations above 30 ng m$^{-3}$ or $N_{10}$ number concentrations greater than 5000 cm$^{-3}$. Due to the position of the sampling inlet at the bow of the ship, data were also excluded where wind directions were from the rear of the ship, defined as between 120–240° relative to the ship's heading.

### 3.2 Particle diameter mode fits

A modified version of the lognormal mode fitting scheme discussed in Modini et al. (2015) was applied to the particle size distributions. The SMPS size distributions were averaged to a 2-hour time base and up to nine lognormal modes were fitted to each distribution using the "Mclust" finite mixture modelling package in the R software environment (Scrucca et al., 2016). This process used the Bayesian Information Criterion, an estimator of model accuracy that penalises the complexity of the model to avoid overfitting. These modes were then grouped under two particle size distributions modes, representing Aitken and accumulation mode aerosol, respectively. The fitted modes were allocated according to their position above or below the Hoppel minimum, defined as the diameter bin corresponding to the minimum number concentration between 40–100 nm. To aid in selecting the most representative local minimum within this range, the SMPS distributions were smoothed using a rolling average across the diameter bins, with a 6-bin averaging window. Representative modal diameters for the aggregated Aitken and accumulation distributions were calculated from their respective geometric means. Since the distributions were limited to a maximum diameter of 660 nm, there were insufficient diameter bins above 500 nm to adequately constrain an additional primary marine aerosol mode. As such, it should be noted that primary marine aerosol contributes some small but undetermined number fraction towards these distributions.



### 3.3 Classification of air masses

There were few instances of "steady-state" conditions in which the aerosol properties remained largely unchanged over extended periods of time. However, on several instances, the aerosol size distribution became significantly biased towards either the Aitken or accumulation mode and the modal diameters abruptly shifted. The rapid onset of these changes suggested

a change in air mass origins. The likely sources of these distinct air masses or aerosol production sources were investigated by examining the corresponding air mass back trajectories, radon concentrations and the associated aerosol chemical composition.

### 3.4 Rasterization of productive region

The maps of oceanic Chl-a indicated numerous areas of elevated biological productivity during the austral summer. Aside from the nutrient-rich Antarctic coastal waters, the highest concentrations were seen in a broad region stretching southwards

from the Kerguelen Plateau to Antarctica (Fig. 1). Due to the prevailing direction of air masses, this region was viewed as a likely source for precursors of NPF and modification of existing aerosol populations. The 4 km resolution of the Chl-a concentrations was beyond the spatial accuracy of the air mass paths estimated with HYSPLIT. To aid identification of air masses which had a higher likelihood of being influenced by biological activity, the productive Kerguelen-Antarctic region was subdivided into a grid of 3 x 6 cells, using an equirectangular projection. Chl-a concentrations were averaged within each

cell. In order to specifically target high Chl-a concentrations, a threshold was applied to disregard any cells with monthly-average concentrations lower than 150 % of the mean for the whole rasterized region. The southern-most row of cells substantially overlaps the Antarctic coastline and is therefore only relevant to air masses that pass over a comparatively narrow band of ocean. This was intentionally chosen to ensure that the highly productive Prydz Bay was included, and to provide latitudinal constraints for the high concentrations along the coastline, reducing their bias on the typically lower concentrations

seen further from the coast.

The gridded and thresholded region was subsequently used to calculate the proportion of each back trajectory that passed through highly productive cells while within the marine boundary layer. A proxy for the cumulative exposure to biogenic emissions was also estimated from the product of the time that an air mass back trajectory spent in the MBL over each productive cell and the corresponding cell-averaged Chl-a concentration.

## 4 Results and discussion

### 4.1 General observations

Relatively warm conditions were experienced throughout the voyage with minimum air and sea surface temperatures of -2.7 and 0.8 °C, respectively. At the ship, mean wind speeds were 11.6 m s$^{-1}$ and frequent synoptic-scale weather systems were observed passing the ship throughout the voyage, particularly at latitudes greater than 50° S.



A time series of particle size distributions is shown in Fig. 2(a). The Aitken mode aerosol represented the greater proportion of the total aerosol number throughout most of the voyage, with a geometric mean diameter frequently as low as 20 nm. These periods are attributed to maritime Southern Ocean (*mSO*) air masses, with typically long fetches over the open ocean to the south west and radon concentrations which averaged $43 \pm 17$ mBq m$^{-3}$, in good agreement with the Southern Ocean

background concentration of 50 mBq m$^{-3}$ given in Chambers et al. (2018).

The long south-westerly fetches were frequently disrupted by the passage of synoptic and mesoscale weather systems. This was notably observed on 4$^{th}$ – 6$^{th}$, 8$^{th}$ – 9$^{th}$ and on 11$^{th}$ February. On all three occasions, the aerosol number concentrations decreased, the size distributions became biased towards the accumulation mode and radon concentrations increased above the marine background (Fig. 2). Air mass trajectories and radon concentrations indicated that the first period was influenced by

continental and coastal Australian air masses (*cAU*), while the latter two received air masses from continental Antarctica (*cAA*). The mean aerosol properties observed from each air mass are shown in Table 1. They are further subdivided into discrete episodes for each air mass in Table S1 and are discussed in detail in the following sections.

## 4.2 Characteristics of the *mSO* air mass

Across the four *mSO* sampling periods, the mean N$_{10}$ concentration of 541 cm$^{-3}$ was almost double the concentrations observed

from the continental air masses. This difference was driven by a substantially higher Aitken fraction, representing up to 94 % of the total number concentration, as shown in Fig. 3. With modal diameters averaging 30 nm, the Aitken aerosol were often smaller than those seen in the continental air masses (Fig. S2) and both their concentration and modal distributions were highly variable, rarely remaining constant for more than a few hours at a time.

These findings are in good agreement with observations reported by Fossum et al. (2018), from a concurrent cruise in the

Atlantic sector of the Southern Ocean, although they observed Aitken aerosol with a slightly larger mode diameter of 42 nm. Likewise, Schmale et al. (2019) reported similar total aerosol number concentrations. The relative change in number concentration between measurements in the mid-latitudes of the Southern Ocean and those taken close to the Antarctic continent during their ACE-SPACE circumnavigation of Antarctica was also similar to our findings.

An additional smaller Aitken mode centred at approximately 10–20 nm was seen on multiple occasions during these *mSO*

periods. It was particularly obvious on 3$^{rd}$, 6$^{th}$, 9$^{th}$ and 11$^{th}$ November (Fig. 2) and suggested that NPF had occurred, but that the precursor concentrations or atmospheric conditions had not supported sufficient particle growth to reach diameters comparable to the remainder of the Aitken distribution. The NAIS showed no evidence of aerosol formation or growth of sub-10 nm particles to these sizes, indicating that the NPF events occurred sometime prior to the air mass reaching the ship.

Episodes of increased Aitken mode number concentrations were most common when air masses maintained a long fetch over

the productive region between the Kerguelen Plateau and the Antarctic coastline (Fig. 4(a) and Fig. S1). This was a common occurrence, particularly for *mSO2* and subsequent air masses, and suggests that the Aitken mode observed during these periods is significantly influenced by the biological productivity in that region, leading to NPF from biogenic precursors. However,



this characteristic back trajectory path alone does not fully explain the variability in Aitken concentrations and diameters. Specifically, there was no consistent correlation between the Chl-a exposure proxy parameter and any Aitken mode parameter, such as number concentration or mean diameter. Therefore, we have chosen to examine these periods in more detail to identify other contributing influences.

### 4.2.1 *mSO4*

Shortly after the transition to the *mSO4* sampling period, the $N_{10}$ concentrations peaked at 800 cm$^{-3}$, due to an abrupt increase in Aitken mode aerosol, with geometric mean diameters that decreased to approximately 22 nm (Fig. S2). As shown in Fig. 4(a), the initial hours of *mSO4* were also associated with a sudden increase in the Chl-a exposure proxy parameter, indicative of the amount of time that the air masses spent within the MBL over the Kerguelen-Antarctic productive region and the corresponding Chl-a concentration along their trajectories. As such, the air masses may have been exposed to an increased flux of biogenic precursors. Furthermore, the trajectories exhibited a short excursion into the free troposphere that could have triggered NPF, resulting in the smaller aerosol mode initially observed during *mSO4*.

After approximately 14 hours, the Aitken number concentrations briefly dropped to the same level as the accumulation mode aerosol despite Chl-a exposures higher than at any other point during the voyage. SSA mass concentrations demonstrated a strong peak during this time (Fig. S3) and there was a general decrease in the altitude of the back trajectories, with less time spent in the free troposphere. Together these observations likely explain the diminished Aitken mode, with weakened vertical transport inhibiting secondary aerosol formation and depletion of condensable material due to the high surface area presented by SSA.

The Aitken mode then returned to even higher concentrations than observed at the start of the *mSO4* period despite back trajectories remaining within the MBL. Both SSA mass concentrations and accumulation mode number concentrations rapidly decreased, while total rainfall progressively increased along the air mass trajectories (Fig. S4), suggesting that these condensation sinks must have been sufficiently depleted to permit NPF to occur within the MBL.

### 4.2.2 *mSO3*

A strong increase in Aitken number concentrations was also observed midway through the *mSO3* sampling period (Fig. 4(b)). This coincided with air mass trajectories with a long fetch over the productive region, followed by a brief excursion into the free troposphere as they passed through a cold front associated with a strong low-pressure system. As seen for air masses early in *mSO4*, the assumed increased flux of precursors followed by vertical transport likely provided the necessary conditions for substantial Aitken mode production. It should be noted that the initial 16 hours of *mSO3* exhibited much higher Chl-a exposures and similar levels of vertical transport, however there was much less impact on the Aitken mode. Again, SSA concentrations were particularly high during these initial hours (Fig. S3) and may have provided a greater condensation sink inhibiting NPF and growth into the Aitken mode.



### 4.2.3 *mSO2*

The *mSO2* air mass exhibited an initial burst of Aitken mode aerosol with a mean diameter of 20 nm. This was similarly associated with a strong frontal system which drove the air mass into the free troposphere and was accompanied by increased rainfall, leading to depletion of the accumulation mode. The vertical transport and subsequent entrainment into the MBL occurred approximately 12 hours prior to the air mass reaching the ship, allowing minimal mixing or dilution time within the MBL and leading to the intense and well-defined mode displayed in Fig. 2. Similar conditions persisted for most of the day, with trajectories over the same approximate regions and vertical transport close to the ship, however the rainfall somewhat decreased. Within three hours, the accumulation mode returned, and the 20 nm mode was almost entirely replaced by an Aitken mode with the largest mean diameters seen throughout the voyage. The large Aitken mode diameters could be due to the transition to an aged air mass which had time for growth via coagulation and/or from condensation onto an existing Aitken population, significantly increasing its size. Subsequently, rainfall rates almost doubled, once again leading to depletion of both the accumulation mode and the larger Aitken mode diameters.

Each of these three *mSO* periods presents evidence of the apparent role of synoptic-scale weather systems in promoting NPF and growth of the Aitken mode by providing convective transport into the free troposphere and depletion of competing condensation sinks through rainfall. However, the high wind speeds associated with these weather systems can also lead to SSA production. During *mSO3* and *mSO4*, high SSA concentrations seem to have increased the condensation sink and inhibited NPF. In contrast, SSA concentrations did not substantially increase during mSO2 despite back trajectories which passed through a cold front. Other studies have discussed significant uncertainties in modelling the SSA flux, with dependence on water temperature, biological activity and other parameters (Saliba et al., 2019; Grythe et al., 2014). The apparent impact of SSA on precursor availability for NPF and secondary nss-SO4 formation highlights the importance of improving this SSA flux parameterisation.

### 4.3 Continental and coastal Australian air masses (*cAU*)

As the ship moved southward of 54.2° S, the aerosol size distribution became dominated by the accumulation mode (Fig. 5), while both aerosol modes exhibited the largest mean diameters of the voyage and $N_{10}$ concentrations remained persistently low (Table 1 and Fig. S2). Radon, BC and organic aerosol concentrations initially rose above the marine background level (Fig. 2(b) and Fig. S3), peaking at hourly means of 220 mBq m$^{-3}$, 0.027 µg m$^{-3}$ and 0.18 µg m$^{-3}$, respectively. The interpretation of this elevated radon concentration has been discussed in detail by Chambers et al. (2018) and it suggests that the air mass was influenced by terrestrial emissions. The back trajectories indicated air masses circulating around a persistent high-pressure system which was centred approximately 1000 km south of the Australian mainland (Fig. 6). Within the 7-day extent of each back trajectory, there was little to no passage over the Australian continent itself, however continental species could have persisted from prior passage over the continent because the air masses measured during this period were in the free troposphere for greater than 50% of the time. As a result of this bias towards large diameters, and despite the relatively high



organic loading, 79 % of the total aerosol number activated as $CCN_{0.5\%}$. This high activation ratio was particularly demonstrated during the afternoon and evening of 4th February, when rainfall in the vicinity of the ship heavily depleted the aerosol load, with total number concentrations as low as 63.9 cm$^{-3}$.

Over the following day (5 February 2015), as the ship moved further southwards from the Australian continent, radon and BC

concentrations diminished to near background levels (60 mBq m$^{-3}$ and 9 ng m$^{-3}$, respectively). However, other observations did not reflect a return to the usual maritime sampling conditions. There was little variation to the aerosol size distribution and back trajectory paths. Organic aerosol concentrations progressively fell below the detection limit, but SSA also remained low. MSA and non-MSA sulfates rose to peak hourly-average concentrations of 0.035 ug m$^{-3}$ and 0.316 ug m$^{-3}$, the highest values for the voyage and the $CCN_{0.5\%}$ activation ratio reached a campaign maximum of 93 %. As shown in Fig. 1, a wide band of

elevated biological activity was present in the waters south of the Australian coastline. The high-pressure synoptic system had persisted in this region for at least 5 days prior to the air masses reaching the ship, deflecting cold fronts and their associated high wind speeds and rainfall. High biological activity, a relatively low SSA condensation sink and potential for cloud processing created conditions for the growth of large accumulation-mode nss-SO$_4$, rather than conditions promoting the high concentrations of Aitken-mode aerosol typically observed from the *mSO* air masses.

### 4.4 Continental Antarctic air masses (*cAA*)

Each *cAA* period was preceded by strong low-pressure systems centred at approximately 60° S and advancing eastwards towards the ship. As these low-pressure systems advanced, the back trajectories diverted southwards, bringing air masses from the Antarctic continent (Fig. 7 and Fig. 8). The first of these periods (*cAA1*) was encountered within 400 km of the Antarctic coastline, while the ship was located between latitudes of 64–65.1° S. While over the continent, the air masses were almost

constantly within the free troposphere, typically dropping into the MBL after leaving the coastal region, similar to what was the dominant trajectory regime observed in East Antarctica during the austral spring of 2012 (Humphries et al., 2016). The continental influence was supported by mean radon concentrations of $82 \pm 25$ mBq m$^{-3}$, consistent with the increase in summer-autumn radon concentrations at latitudes above 64° S reported by Chambers et al. (2018) through a combination of increased coastal emissions (due to exposed rocks and soils) and subsidence of terrestrially-influenced tropospheric air over

Antarctica. The measurements from approximately half of this 37-hour period were affected by ship exhaust and have been excluded. In the remaining measurements, BC mostly remained at levels comparable to the *mSO* periods, with the exception of the final four hours during which time BC briefly peaked at 0.024 ug m$^{-3}$. As indicated by low $N_{10}$ concentrations and favourable wind conditions, these latter hours of sampling were not characteristic of direct ship emissions, suggesting that the elevated BC may have been from Antarctic terrestrial sources or residual ship exhaust emitted during the southward leg of the

voyage. Alternatively, Chambers et al. (2017) found evidence for air masses influenced by mid-to-low latitude land masses which were transported through the free troposphere, subsided over Antarctica and were subsequently exported to the MBL





via katabatic outflows. As such, the radon and BC concentrations observed during *cAA1* may include residual continental emissions from Australia or New Zealand, present in air masses which subsided over the Antarctic continent.

As with the *cAU* air mass, $N_{10}$ concentrations were well below the voyage mean and the accumulation mode represented 70 % of the total number concentration (Fig. 9), in close agreement with *cAA* observations given by (Fossum et al., 2018). The

additional small mode seen at approximately 10 nm in Fig. 9 can be attributed to a brief period during *cAA2* in which a tri-modal distribution was observed and will be discussed in further detail below.

During *cAA1*, the large contribution from the accumulation mode drove $CCN_{0.5\%}$ concentrations to their highest level with a mean of 240 cm$^{-3}$ and a $CCN_{0.5\%}$ activation ratio of 0.79. The transitions between *mSO* and *cAA1* air masses were well defined during both the southward and northward transits, with pronounced and persistent changes particularly in $N_{10}$ and Aitken

number concentrations, as well as meteorological parameters such as air temperature and absolute humidity (Fig. S5). They occurred across a latitude range of approximately 64.0–64.5° S. During a nearby cruise in the early austral spring of 2012 and in the same region, Humphries et al. (2016) observed a similar abrupt transition at 64.4° S and attributed this to a shift between the Ferrel atmospheric cell and the polar cell. It is interesting to note that while they reported comparable $N_{10}$ concentrations in the polar region, these decreased to a mean of 194 cm$^{-3}$ at lower latitudes, rather than increasing as was seen during the Cold

Water Trial voyage reported on here. This may reflect greater seasonal variability in the Southern Ocean compared to polar waters, with lower oceanic productivity in the early spring leading to reduced rates of aerosol formation and growth to diameters above 10 nm. Nonetheless it is difficult to draw conclusions from these two datasets in isolation, highlighting the need for further latitudinal and seasonal studies in this region.

The *cAA1* period was also characterised by the lowest mass concentrations of SSA and as a result, nss-$SO_4$ represented 72.5 %

of the sub-micron aerosol composition. Particularly at 40 and 150 nm, the aerosol exhibited less volatility than almost any other time during the voyage, suggesting minimal contribution from sulfuric acid. Given the presence of seal and penguin colonies on the Antarctic coastline, the low volatility may reflect the neutralising effect of these biogenic sources of nitrogen to air masses passing over the coastline (Legrand et al., 1998), transforming sulfuric acid into ammonium sulfate and bisulfate. Again, these aerosol properties were generally consistent with the concurrent measurements of *cAA* air masses off the coast of

West Antarctica, given by Fossum et al. (2018). The only significant exception was the contribution from MSA which could not be adequately quantified at such low aerosol concentrations due to the previously discussed instrument limitations. Regardless, since *cAA1* air masses persistently maintained high altitudes while over the landmass, the similarity between observations on opposite sides of the continent may reflect a degree of uniformity in the aerosol populations throughout the Antarctic free troposphere at this time of year.

The second period of *cAA* air masses (*cAA2*) was observed over a 17-hour period while the ship was between 57.5–60.1° S, approximately 1000 km from the Antarctic coast. Compared to *cAA1*, the back trajectories spent less time in the free troposphere, except during the first few hours when a cold front passed through the region. The transitions between the *mSO* and *cAA2* air masses were more gradual, with a greater proportion of each back trajectory passing over the coastal region (Fig. 8), particularly during the four hours at the start and end of this period. Radon concentrations were only slightly elevated



above the marine background, at a mean of 62.5 mBq m$^{-3}$, but the period otherwise shared the same mean N$_{10}$ concentration and nss-SO$_4$ mass concentration as *cAA1* (Table S1, Fig. S2 and Fig. S3).

The accumulation mode retained a similar mean diameter to *cAA1* at 122 nm, however its contribution to the total size distribution decreased to 44 %. In addition, a distinct tri-modal distribution developed during the first six hours of *cAA2*, with

an additional small Aitken mode centred at approximately 15 nm (Fig. 2). The corresponding back trajectories spent increased time over the highly productive coastal region, which may have offered more opportunities for uptake of biogenic sulfate precursors. Likewise, the passing weather system may have provided sufficient uplift to spur NPF. If so, it seems that the meteorological conditions or precursor concentrations were insufficient to support further growth into the Aitken size range commonly seen in the *cAA* air masses.

Another marked difference between the two *cAA* periods can be seen in the elevated SSA mass concentrations during *cAA2*. As highlighted above, both high SSA concentrations and the presence of small diameter Aitken mode aerosol were characteristic of the *mSO* sampling periods. Given the relatively low radon concentrations during *cAA2*, substantial dilution and mixing likely occurred over the longer transport range. Therefore, these two *cAA* periods may demonstrate the progressive transition of air masses from being strongly influenced by continental outflow into characteristic maritime air masses as they

are transported further from the continent.

## 4.5 Latitudinal trends

The north-south track from coastal Australian to polar Antarctic waters provided an ideal opportunity to examine trends in aerosol properties across the latitudinal range of the Southern Ocean (Fig. S6 and Fig. S7). Simple linear regression with respect to latitude indicated an approximate 25% decrease in the mean diameter of the Aitken mode between 46–63° S,

however the accumulation mode did not exhibit a significant trend. In contrast, the hygroscopicity of 150 nm aerosol increased from 0.40 to 0.59 between 51–63° S, while there was no significant change at 40 nm. The increased density of intense weather systems at high latitudes would have increased scavenging rates for accumulation mode aerosol while promoting an increased SSA population. In turn, this likely lead to the reduction in aerosol diameter and an SSA-driven increase in hygroscopicity at larger diameters without substantially affecting the 40 nm population.

As previously discussed, N$_{10}$, Aitken, accumulation and CCN number concentrations changed substantially within the polar cell, south of 64° S. With the exception of this region, and despite the above trends in diameter and hygroscopicity, the concentrations did not exhibit any clear latitudinal dependence over the breadth of the Southern Ocean (Fig. S8 and Fig. S9). Instead they showed high levels of variability, particularly south of 48° S, driven by short-term changes in air mass and the impact of synoptic events on aerosol production and scavenging rates. For example, at 55° S, there was a pronounced decrease

in both CCN and N$_{10}$ concentrations. Although this occurred in both the southward and northward transits, it is unlikely to be a consistent latitudinal feature since it can be attributed to transient events including aerosol losses during the *cAU* rain event and the brief episode of high SSA mass concentrations coinciding with Aitken mode depletion during *mSO4*.





## 5 Conclusions

We have presented the first comprehensive study of aerosol properties targeting the full latitudinal width of the Southern Ocean south of Australia. The voyage was carried out during the austral summer and extended to the edge of the marginal ice zone at 65° S. Heightened summer-time marine productivity was clearly apparent with strong Chl-a concentrations along the

Antarctic coastline and extending from the Kerguelen Plateau towards Antarctica. On multiple occasions, the Aitken mode number fraction increased to 75 % of the total aerosol population in air masses passing over the Kerguelen-Antarctic productive region. However, this was typically associated with relatively small Aitken mode geometric mean diameters, at approximately 30 nm. As a result, the number concentration of Aitken mode aerosol correlated poorly with CCN concentrations throughout most of the voyage. Assuming that this is a true reflection of typical aerosol populations in this region of the Southern Ocean,

it suggests that nss-$SO_4$ NPF from this region do not have a direct impact on the local cloud droplet number concentration and that further cloud processing or nss-$SO_4$ condensation is required to grow them to cloud-active diameters.

CCN concentrations were more significantly impacted by changes in air mass, with much higher CCN ratios observed in aerosol transported from the Antarctic and Australian continents. These episodes were observed at distances of over 1000 km from either continent, demonstrating that long range transport of continental air masses can effectively influence the full width

of the Southern Ocean, as previously proposed by Chambers et al. (2018). Furthermore, it is interesting to note that two episodes were observed during this 15-day dataset, which suggests that there could be an important contribution to the Southern Ocean from aerosol transported from the Antarctic and Australian continents.

The changes in air mass were driven by synoptic-scale weather systems. The prolonged transport of continental and coastal Australian aerosol was maintained by a persistent, slow-moving high-pressure system. Similarly, during the *cAA2* period, a

series of low-pressure systems drew the Antarctic air masses deep into the Southern Ocean. During *mSO3* and *mSO4*, increased generation of SSA inhibited the formation and growth of the Aitken mode, while in *mSO2*, heavy rainfall and depletion of larger-diameter particles was associated with an intense burst of Aitken mode aerosol. Finally, aside for the latter part of *mSO4*, enhancement of the Aitken population was consistently associated with vertical transport between the MBL and the free troposphere during the passage of cold fronts. Clearly, synoptic systems play a fundamental role in mediating aerosol properties

in the Southern Ocean through their impact on air mass origins and trajectories, SSA production, aerosol removal rates due to rainfall and vertical transport between the MBL and the free troposphere.

In line with observations by Humphries et al. (2016), a pronounced change in aerosol number concentrations, size distributions, CCN activity, air mass trajectories and other meteorological parameters offer support for a transition into the polar atmospheric cell at an approximate latitude of 64° S. The highest CCN concentrations were observed within this region, despite low $N_{10}$

concentrations, with air masses primarily sourced from the free troposphere over the Antarctic continent. In contrast, most aerosol properties exhibited little dependence on latitude throughout the remainder of the voyage. Instead, austral summer influences from marine productivity, long range transport, and synoptic weather systems resulted in a high degree of variability which outweighed most latitudinal trends.



Given the current scarcity of data from the Southern Ocean, and the over-representation of summer-time voyages, there is a clear need for a greater body of data. This would allow examination of whether the aerosol population exhibits such variability during other seasons and therefore its dependence on biological activity. It would also be valuable to assess the frequency, duration and characteristics of the long-range transport events.

## 6 Data availability

The underlying research data can be accessed upon request to the corresponding author (Zoran Ristovski; z.ristovski@qut.edu.au).

## 7 Author contribution

ZDR was chief scientist for the Cold Water Trial campaign and coordinated the investigation in conjunction with MDK. ZDR, LTC and JA managed data collection, daily maintenance and calibration of the aerosol instrumentation throughout the voyage. Installation, calibration and data analysis for the ACSM was performed by PS. Processing and interpretation of Radon-222 data was completed by SDC and AGW. All other data analysis and interpretation was led by JA with input from all authors, particularly LTC and ZDR. JA led manuscript preparation with input from all authors.

## 8 Acknowledgements

This work was funded by an Australian Government Research Training Program Scholarship and an ARC Discovery grant (DP150101649). We acknowledge the NASA Goddard Space Flight Centre, Ocean Biology Processing Group for the provision of the MODIS Aqua chlorophyll data and the NOAA Air Resources Laboratory (ARL) for the provision and support for the HYSPLIT transport and dispersion model. The Authors wish to thank the CSIRO Marine National Facility (MNF) for its support in the form of sea time on RV Investigator, support personnel, scientific equipment and data management. In particular, we thank the technical and IT support personnel on board the voyage, including Ian McRobert, William Ponsonby, Brett Muir, Steve Thomas, Hugh Barker, Stewart Wilde and Anoosh Sarraf. We gratefully acknowledge Jason Ward and James Harnwell (CSIRO) for their ongoing support of the permanent aerosol instrumentation on board. All data and samples acquired on the voyage are made publically available in accordance with MNF Policy from the CSIRO Data Access Portal (https://data.csiro.au/dap/).



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


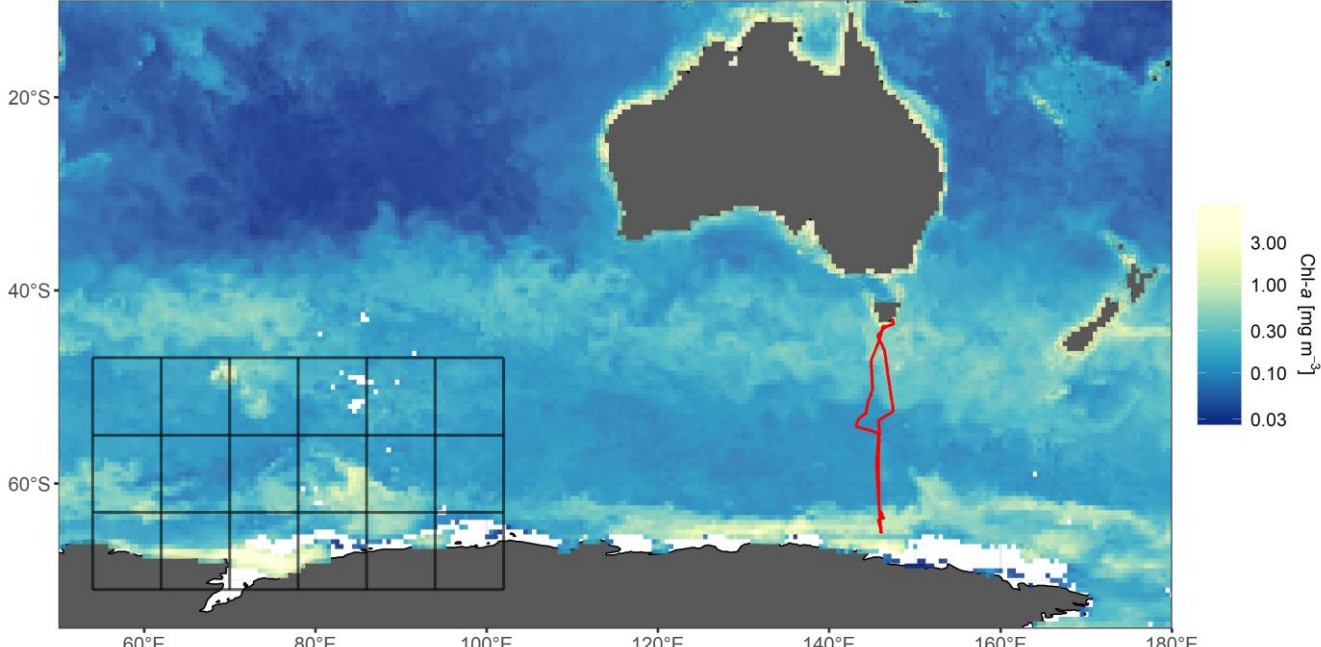

**Figure 1: Mean chlorophyll-a concentrations for February 2015, derived from MODIS satellite images. A logarithmic colour scale has been used and has been limited to the range 0.03–7 mg m$^{-3}$ to enhance visual detail. White regions represent unavailable data due to sea ice or persistent cloud cover. The black grid represents the cell boundaries used to rasterize Chl-a concentrations in the Kerguelen-Antarctic productive region. The ship's voyage track is represented in red.**


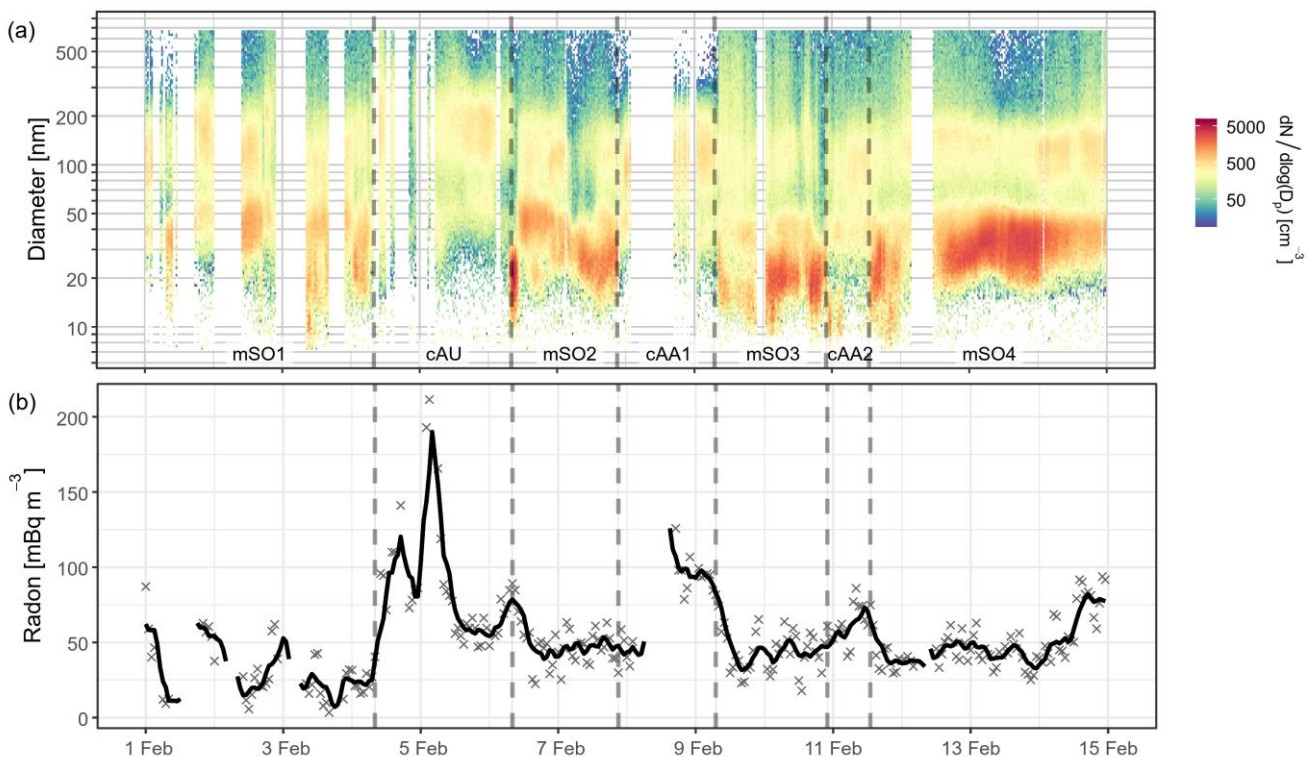

**Figure 2.** Time series of (a) aerosol size distributions and (b) atmospheric radon concentrations throughout the Cold Water Trial voyage. Periods influenced by different air masses are labelled and delimited by dotted lines. Blank regions represent periods of instrument maintenance or contamination from ship emissions which have been excluded from analysis. The smoothed black line represents a rolling six-hour average of the radon concentrations to assist in identification of temporal trends.




**Table 1: Mean physical properties observed for the three air mass classifications, presented as the mean for each air mass ± the standard deviation.**

| Air mass | *cAA* | *cAU* | *mSO* |
|---|---|---|---|
| **Number concentrations** | | | |
| $N_{10}$ (cm$^{-3}$) | 310 ± 37 | 220 ± 70 | 540 ± 200 |
| $CCN_{0.5\%}$ (cm$^{-3}$) | 210 ± 45 | 180 ± 65 | 190 ± 65 |
| $CCN_{0.5\%}$ activation ratio | 0.68 ± 0.17 | 0.79 ± 0.11 | 0.35 ± 0.18 |
| **Size distributions** | | | |
| Aitken number fraction | 0.40 ± 0.16 | 0.40 ± 0.07 | 0.71 ± 0.14 |
| Aitken mean diameter (nm) | 31 ± 9 | 43 ± 11 | 30 ± 6 |
| Accumulation mean diameter (nm) | 115 ± 14 | 191 ± 18 | 137 ± 24 |
| Hoppel minimum diameter (nm) | 57 ± 7 | 76 ± 7 | 69 ± 8 |
| **Hygroscopicity** | | | |
| $\kappa_{D40}$ | 0.41 ± 0.10 | 0.38 ± 0.09 | 0.38 ± 0.08 |
| $\kappa_{D100}$ | 0.39 ± 0.08 | 0.37 ± 0.04 | 0.44 ± 0.08 |
| $\kappa_{D150}$ | 0.51 ± 0.12 | 0.46 ± 0.07 | 0.52 ± 0.12 |
| **Volatility** | | | |
| $VFR_{D40}$ | 0.97 ± 0.09 | 0.89 ± 0.03 | 0.89 ± 0.11 |
| $VFR_{D100}$ | 0.94 ± 0.07 | 0.85 ± 0.03 | 0.90 ± 0.07 |
| $VFR_{D150}$ | 0.94 ± 0.08 | 0.92 ± 0.02 | 0.91 ± 0.07 |
| **Composition** | | | |
| Org (µg m$^{-3}$) | BDL[1] | 0.08 ± 0.05 | BDL |
| $SO_4$ (µg m$^{-3}$) | 0.12 ± 0.03 | 0.18 ± 0.08 | 0.12 ± 0.06 |
| SSA (µg m$^{-3}$) | 0.08 ± 0.07 | 0.12 ± 0.10 | 0.21 ± 0.18 |
| **Continental / anthropogenic influences** | | | |
| Radon (mBq m$^{-3}$) | 73 ± 22 | 83 ± 38 | 43 ± 17 |
| BC (ng m$^{-3}$) | 4 ± 5 | 12 ± 6 | 5 ± 6 |

[1] *Below detection limit*



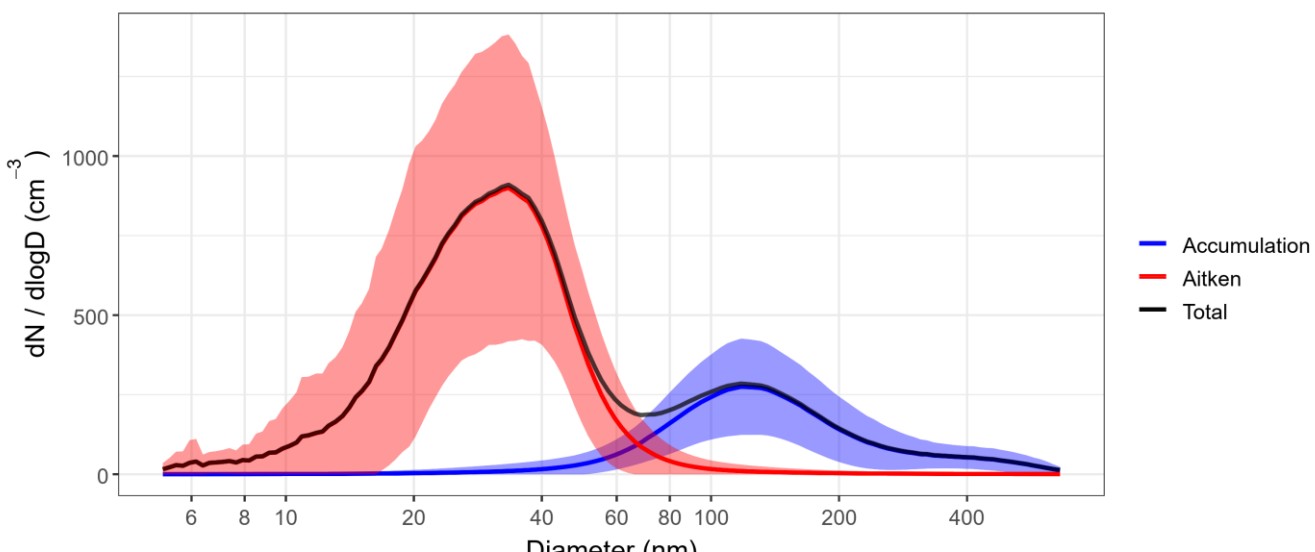

**Figure 3: Mean number size distribution for sub-micron aerosol during periods of *mSO* air masses. The coloured lines are fitted log-normal distributions representing contributions from Aitken and accumulation mode aerosol to the observed (black) total size distribution. The shaded regions give the standard deviations of the fitted modes throughout the *mSO* sampling periods.**





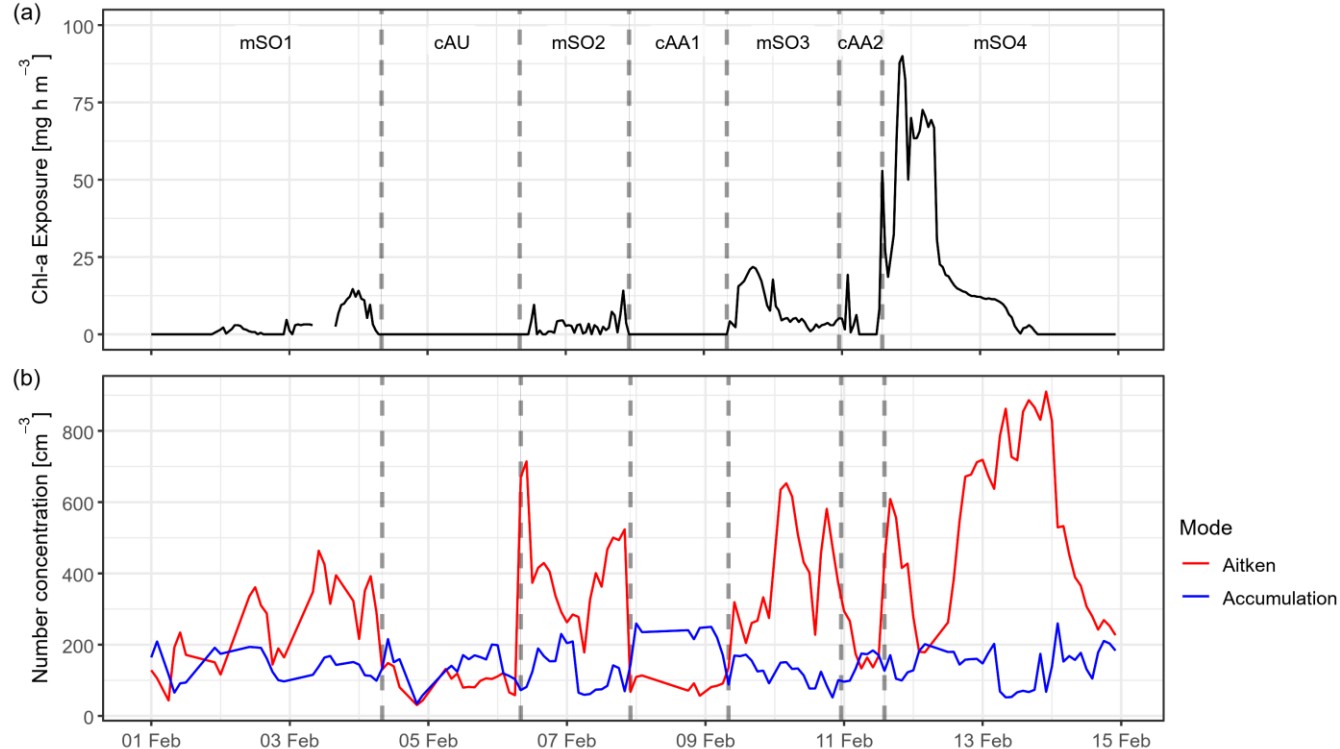

**Figure 4: (a) Summed Chl-a concentration as a proxy for total flux of biogenic aerosol precursors received by each air mass during time spent within the MBL over the productive Kerguelen-Antarctic region. (b) Aerosol number concentrations in the Aitken and accumulation modes, as derived from the fitted size distributions. Periods influenced by different air masses are labelled and delimited by dotted lines.**


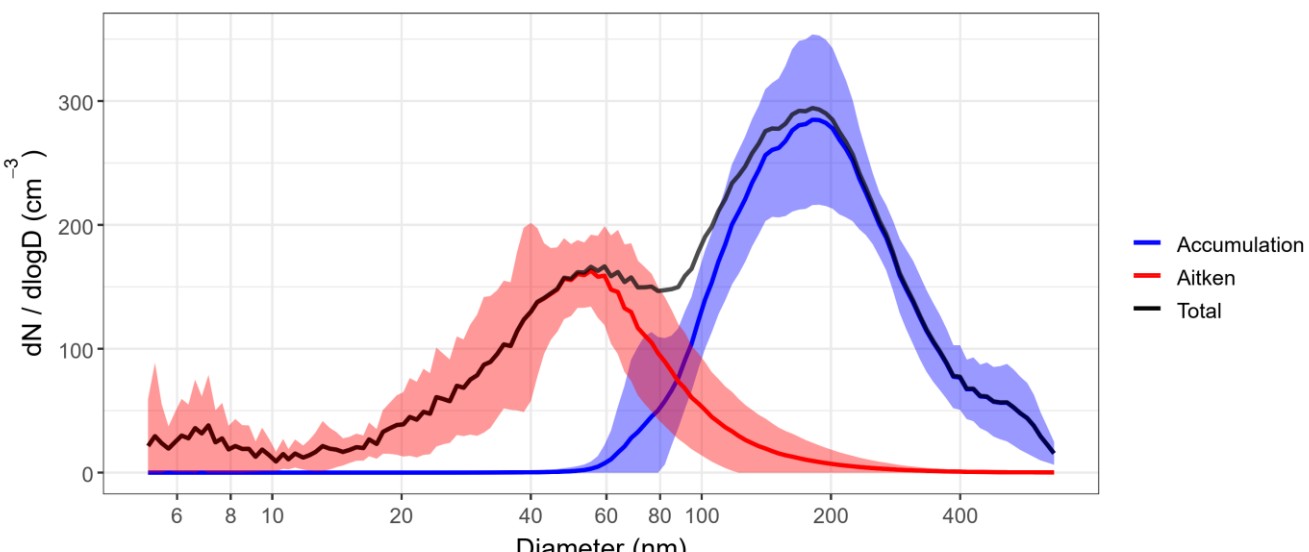

**Figure 5: Mean number size distribution for sub-micron aerosol during a period likely influenced by continental and coastal Australian air masses. The coloured lines are fitted log-normal distributions representing contributions from Aitken and accumulation mode aerosol to the observed (black) total size distribution. The shaded regions give the standard deviations of the fitted modes throughout the *cAU* sampling periods.**

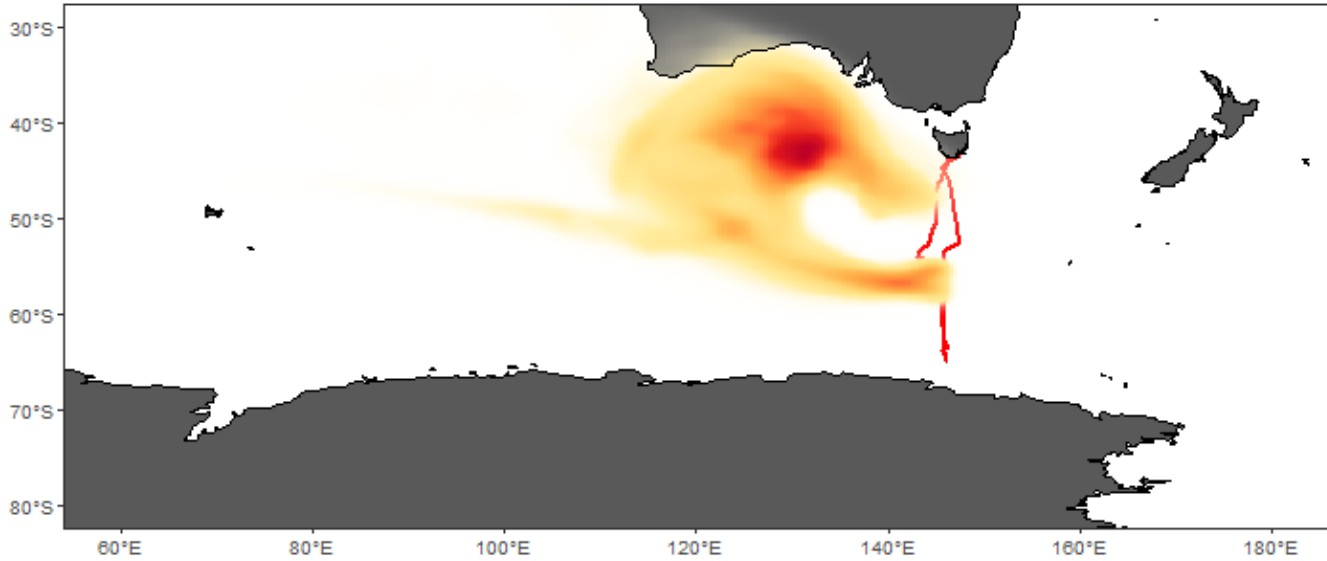

**Figure 6: Probability distribution of air mass coordinates throughout all back trajectory ensembles from the *cAU* sampling period. The colour scale reflects the proportion of trajectories which passed through each location, with the highest density of trajectories found in the dark red region, which coincided with the approximate centre of a high-pressure system. The ship's voyage track is represented in red and the ship was located at a mean latitude of 56° S during this period.**





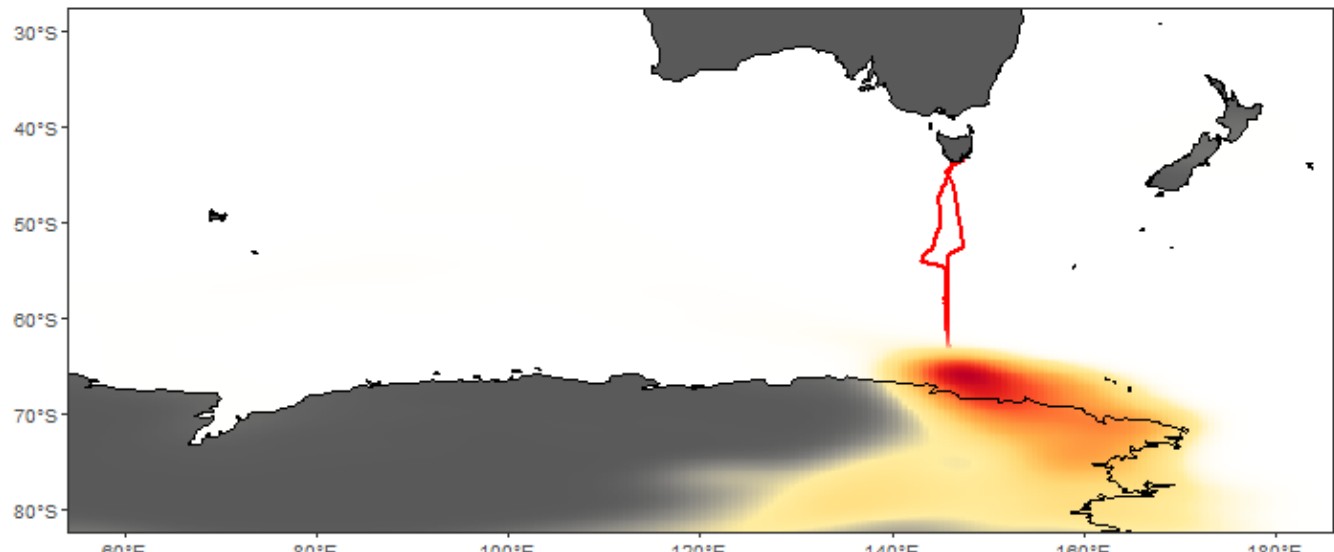

**Figure 7: Probability distribution of air mass coordinates throughout all back trajectory ensembles from the *cAA1* sampling period. The colour scale reflects the proportion of trajectories which passed through each location, with the highest density of trajectories found in the dark red region. The ship's voyage track is represented in red and the ship was located at a mean latitude of 64.5° S during this period.**

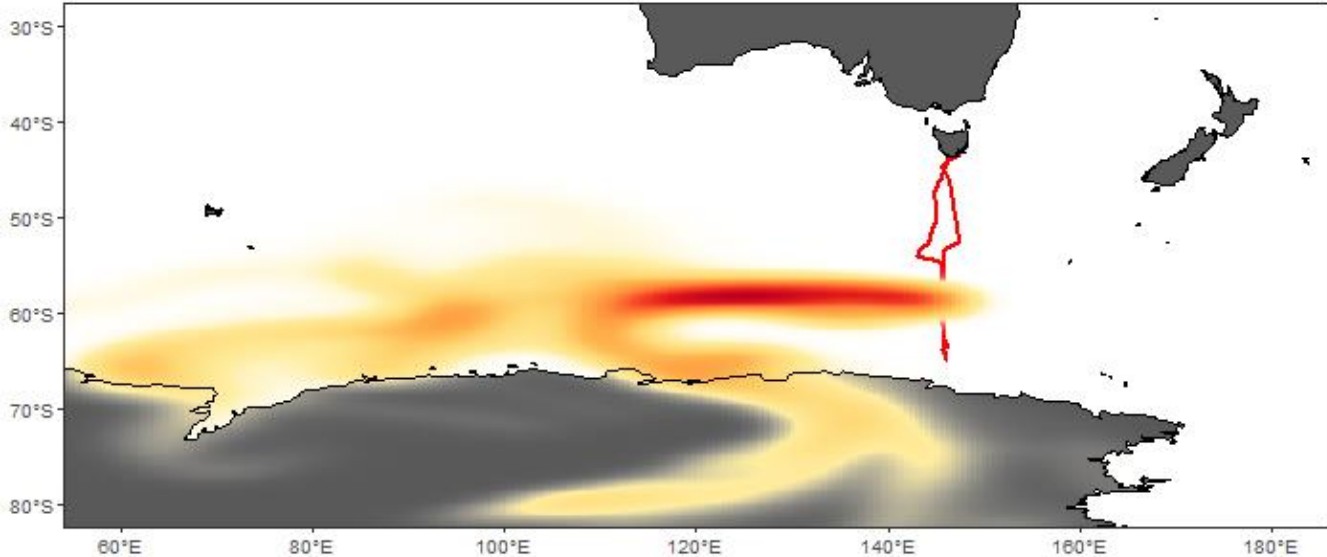

**Figure 8: Probability distribution of air mass coordinates throughout all back trajectory ensembles from the *cAA2* sampling period. The colour scale reflects the proportion of trajectories which passed through each location, with the highest density of trajectories found in the dark red region. The ship's voyage track is represented in red and the ship was located at a mean latitude of 59° S during this period.**





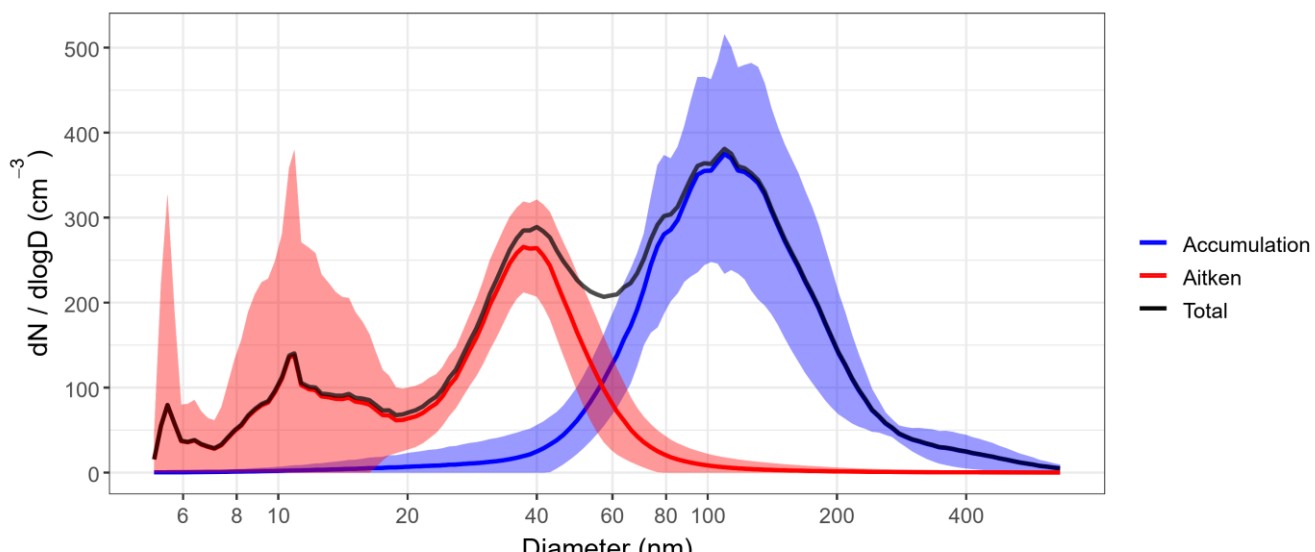

**Figure 9: Mean number size distribution for sub-micron aerosol during two sampling periods influenced by continental Antarctic air masses. The coloured lines are fitted log-normal distributions representing contributions from Aitken and accumulation mode aerosol to the observed (black) total size distribution. The shaded regions give the standard deviations of the fitted modes throughout the *cAA* sampling periods.**