# Peer review of "Marine productivity and synoptic meteorology drive summer-time variability in Southern Ocean aerosols"

_Atmospheric Chemistry and Physics, 2019_

## Referee Comment (RC1) · Matthew Salter (Referee) · 15 Jan 2020

**Summary**

Alroe *et al.* present a set of aerosol and meteorological observations obtained during a three-week test cruise of the *RV Investigator* between Hobart, Australia and the marginal ice zone of Antarctica along longitude 146°. Measurements made during the cruise include aerosol size distributions between 4 and 673 nm, size distributions of ultrafine aerosol (2-42 nm) using a Neutral Cluster and Air Ion Spectrometer (NAIS) instrument, cloud condensation nuclei (CCN) concentration measurements at 0.5%

supersaturation using a CCN counter, aerosol chemical composition measurements using an Aerosol Chemical Speciation Monitor (ACSM), aerosol hygroscopicity and volatility measurements using a Volatility and Hygroscopicity Tandem Differential Mobility Analyser (VH-TDMA), measurements of black carbon mass using a Multi-Angle Absorption Photometer (MAAP) and radon concentrations using a dual-flow-loop two-filter radon detector.

Using this extensive set of observations the authors draw several conclusions:

1. Although Aitken mode number fraction was around 75% on a number of occasions their relatively small median diameter ($\sim 30$ nm) meant that the presence of increased numbers of Aitken mode particles correlated poorly with measured CCN concentrations. This suggests that nss-SO$_4$ new particle formation in the region may have little influence on local cloud droplet number concentrations and that further cloud processing or nss-SO$_4$ condensation is required to grow them to cloud-active diameters.

2. The authors note that CCN concentrations increased in aerosol transported from the Antarctic and Australian continents. This suggests that long-range transport of continental aerosols can effectively influence the entire Southern Ocean south of Australia.

3. As well as influence from the Australian continent, the authors also observed the influence of the Antarctic continent long distances offshore.

4. The authors note the important role the synoptic situation played in mediating aerosol properties during their expedition, especially the role of vertical transport between the marine boundary layer and the free troposphere in enhancing the number of Aitken mode particles.

5. The authors present evidence of a pronounced change in aerosol properties at $\sim 64°$S which they attribute to the transition into the polar atmospheric cell.

**Major points**

Unfortunately, I cannot recommend that this manuscript be accepted for final publication in ACP since in my eyes the scientific significance of the work is quite simply too low. The authors appear to have collected a nice dataset that is nicely presented in a well-written manuscript. However, the authors have essentially not gone beyond describing their measurements. As such, having read the manuscript I was left wondering what I had learned - the conclusions presented above hardly scratch the surface and are essentially well-established. Given this, my recommendation to the authors would be to submit the dataset and accompanying article to a journal for the publication of articles on original research data such as Copernicus's Earth System Science Data journal. If the authors do want to continue to present this research in ACP they need to delve far deeper into interpreting the data and ask themselves what this dataset can contribute that will take the field as a whole forward - in my eyes this goes beyond major revisions.

**Minor points**

Page 4, line 5 - In my eyes stating that "the ship and its sampling facilities are discussed in detail" elsewhere is not particularly helpful. I would like to see at least the basic information presented here alongside the data.

Page 6, line 29 - ". . . number fraction towards these distributions" would read better as "number fraction towards these sizes".

Page 14 - Although this is perhaps more personal taste, in my view the conclusions section is rather more of a summary. The conclusions of the paper should be more concise than its current form.

---

## Referee Comment (RC2) · Anonymous Referee #2 · 24 Jan 2020

The paper by Alroe et al. reports the result of a Southern Ocean cruise and marine aerosol features from the underexplored region. The paper is fluently written and was a pleasure to read, but unfortunately contains little scientific advancement beyond the current state of knowledge. The results broadly agree very well with the already published papers in the topical area, but I was wondering why the authors did not come up with unique insights possessing a good dataset and all the relevant instruments. Was the study not from the Southern Ocean and without thorough trajectory and ocean colour analysis I would have difficulty recommending it for publication, but I would like to encourage the authors to take a second look at their results and try to enhance the value of the paper by highlighting certain novel aspects. I hope my comments will

motivate the authors and may help in improving the paper. For example, were there any oceanographic data available like in-situ chlorophyll measurements to infer phytoplankton blooms or more general tracer measurements by e.g. fluorometer? Last, but not least properly fitted size distributions may help to deepen the discussion of particle sources.

Key comments

The authors used a sea salt tracer to estimate sea salt mass, but they could also cross-check with SSA estimated mass and number from recent of earlier sea salt source functions at the observed wind speed. Something more can be done here if proper lab calibrations were not performed for ACSM which would have been very useful given instrumental differences between high resolution AMS used by Ovadnevaite et al. and ToF ACSM used in this study.

It is important to realise that the observed particles and their size distributions are a product of cloud cycling. Or in other words, the observed size distribution is a product, not the source for cloud processing. Indeed, all accumulation mode particles can undergo subsequent cloud cycles, but Aitken mode cannot given their modal diameter of 30nm. Bi-modal size distribution has exactly arisen after cloud cycling. If there were no clouds accumulation mode chemical composition would have been entirely made of primary sea salt. As was noted at the beginning properly fitted size distributions (Comments regarding Figure 3) and the above comments may enhance the discussion.

The authors must redo the fitting of log-normal distributions. The shape of “bi-modal” distribution is clearly suggesting more modes, e.g. nucleation mode of ~10nm and second accumulation mode centred at ~420nm. I do not see symmetrical log-normal modes, only one side of them which makes me wondering what the inventive fitting was applied. Specifically, significant departure of log-normal mode from the observed size distribution is suggesting additional mode(s) together with the potentially excessive geometric spread (sigma) above ~1.5. Consequently, nucleation mode is suggesting

contribution from new particle formation at the coast or open ocean depending on the trajectories. But first, please do the proper fitting.

Minor comments

Page 4. Line 8. I could not find in this paper nor the referenced paper information about the sampling inlet. Was it community sampling duct of certain dimensions with individual instruments sub-sampling from it? What was the total flow and laminar conditions? Was it experiencing a significant drying during air passage through it?

Page 4, line 24. Was the drier used for ACSM to limit the excessive humidity?

Page 6, line 13. BC threshold was set conservatively in general, however, given pristine nature of the Southern Ocean may have been set even lower, especially as it was an hourly average. Also while number concentration criterion is appropriate, number concentration is not a great arbiter when new particle formation events may increase N10 concentration substantially. It would very useful to present BC data in Figure 2.

Page 6, line 28. I am confused with this statement as sea spray usually contributes significant number fraction to submicron aerosol based on several established sea spray source functions. Possibly the authors were meant to say that the supermicron sea spray mode contributed a small number fraction to the observed size distributions which would be true.

Page 7, line 28. Please provide ranges of temperature and wind speed.

Page 8, line 31. Notation is confusing with that of sulfur species. Perhaps is better using upper script notation or mSO-first, mSO-second and so on.

Page 9, line 5. Why mSO4 is preceding mSO3 period and mSO3 preceding mSO2?

Page 10, line 19. There is no mention nor reason why CCN not reported for mSO periods.

Page 11, line 29. The only Antarctic terrestrial sources of BC are the scientific bases.

Page 12, line 21. This conclusion must be supported by ammonium concentration and degree of neutralisation (DON). There is no mention nor reason of the absence of ammonium concentration. ACSM is perfectly capable of measuring ammonium ion.

Figure 2. BC concentration would be extremely useful on a separate scale of radon graph.

Table 1. Air mass notations should be spelt below the Table or in the caption. NH4 is an important species discriminating between different neutralisation degrees depending on continental impact. DON could be presented too. DL should be noted below Table or simply like <0.XX

---

## Author Comment (AC1) · 1 May 2020

**Referee #1**

*Alroe et al. present a set of aerosol and meteorological observations obtained during a three-week test cruise of the RV Investigator between Hobart, Australia and the marginal ice zone of Antarctica along longitude 146°. Measurements made during the cruise include aerosol size distributions between 4 and 673 nm, size distributions of ultrafine aerosol (2-42 nm) using a Neutral Cluster and Air Ion Spectrometer (NAIS) instrument, cloud condensation nuclei (CCN) concentration measurements at 0.5% supersaturation using a CCN counter, aerosol chemical composition measurements using an Aerosol Chemical Speciation Monitor (ACSM), aerosol hygroscopicity and volatility measurements using a Volatility and Hygroscopicity Tandem Differential Mobility Analyser (VH-TDMA), measurements of black carbon mass using a Multi-Angle Absorption Photometer (MAAP) and radon concentrations using a dual-flow-loop two-filter radon detector.*

*Using this extensive set of observations the authors draw several conclusions:*

*1. Although Aitken mode number fraction was around 75% on a number of occasions their relatively small median diameter ($\sim$ 30 nm) meant that the presence of increased numbers of Aitken mode particles correlated poorly with measured CCN concentrations. This suggests that nss-SO 4 new particle formation in the region may have little influence on local cloud droplet number concentrations and that further cloud processing or nss-SO 4 condensation is required to grow them to cloud-active diameters.*

*2. The authors note that CCN concentrations increased in aerosol transported from the Antarctic and Australian continents. This suggests that long-range transport of continental aerosols can effectively influence the entire Southern Ocean south of Australia.*

*3. As well as influence from the Australian continent, the authors also observed the influence of the Antarctic continent long distances offshore.*

*4. The authors note the important role the synoptic situation played in mediating aerosol properties during their expedition, especially the role of vertical transport between the marine boundary layer and the free troposphere in enhancing the number of Aitken mode particles.*

*5. The authors present evidence of a pronounced change in aerosol properties at $\sim$ 64° S which they attribute to the transition into the polar atmospheric cell.*

*Unfortunately, I cannot recommend that this manuscript be accepted for final publication in ACP since in my eyes the scientific significance of the work is quite simply too low. The authors appear to have collected a nice dataset that is nicely presented in a well-written manuscript. However, the authors have essentially not gone beyond describing their measurements. As such, having read the manuscript I was left wondering what I had learned - the conclusions presented above hardly scratch the surface and are essentially well-established. Given this, my recommendation to the authors would be to submit the dataset and accompanying article to a journal for the publication of articles on original research data such as Copernicus's Earth System Science Data journal. If the authors do want to continue to present this research in ACP they need to delve far deeper into interpreting the data and ask themselves what this dataset can contribute that will take the field as a whole forward - in my eyes this goes beyond major revisions.*

Author's Answer

The authors appreciate the reviewer's consideration of this manuscript. On reflection, we agree that the scope of the original manuscript was broad and there were considerable opportunities

for more detailed investigation. In response to both reviewers' comments, we have significantly increased the depth and quality of our analysis.

Changes to the analysis include:
- Improved aerosol size distribution mode fitting, including the separation of a nucleation mode throughout much of the voyage
- Modelling of sea salt mass concentrations using a wind speed-dependent model, to allow evaluation of the scaled sea salt measurements obtained from ACSM measurements, and to assess the validity of the model for the conditions observed during this voyage
- Assessing contributions to CCN and the impact of variations in SSA and Aitken mode aerosol concentrations
- Detailed analysis of back trajectories to identify meteorological conditions associated with the above variations in SSA and Aitken mode aerosol

Subsequent to the above changes, we found that the chlorophyll-a exposure over the rasterized productive region (discussed in the original manuscript) was insufficient to explain observed variations in Aitken and nucleation mode aerosol properties. Instead, we have extensively rewritten the manuscript with a greater focus on synoptic-scale meteorological conditions and their key influence on the variability of CCN concentrations in the Southern Ocean.

This has resulted in a much stronger study that highlights the following key findings:
1) Accurate modelling of sea spray aerosol concentrations must be based on detailed air mass meteorological histories, including parameters such as wind speed, rainfall frequency and intensity, and altitude above sea level relative to the marine boundary layer depth. In a dynamic environment like the Southern Ocean, these parameters must be considered along the trajectory of the air mass rather than at the sampling point, and sea spray aerosol are unlikely to reach equilibrium concentrations until the air mass has experienced favourable conditions (low rainfall and low altitudes) for approximately 48 hours.
2) Several periods were identified when CCN concentrations were significantly influenced by either increased wind-related SSA or large-diameter Aitken mode aerosol. Aitken mode aerosol were to be the most significant of these two sources of CCN, potentially providing up to 56% of CCN, even at high latitudes.
3) Despite the high wind speeds associated with frequent cyclonic systems at high latitudes, these weather systems ultimately limited SSA concentrations due to increased rates of precipitation, in-cloud scavenging and vertical transport of air masses into the free troposphere where they became decoupled from the surface source of sea spray. As a result, the highest SSA concentrations and SSA contributions to CCN were observed at lower latitudes.
4) Similarly, while there was frequent evidence of secondary sulfate formation, growth of these aerosol to CCN-relevant diameters was only observed where air masses transited over biologically productive regions at relatively low altitudes, remained within the MBL for 24-48 hours and then encountered a strong cold front which combined rainfall, compression of the MBL and vertical transport into the free troposphere.

Variability arising from subgrid scale meteorology and aerosol properties is poorly represented in current models (Weigum et al., 2016; Lin et al., 2017). This study now provides a detailed examination of the sources of variability in CCN during several episodes of long range transport, sea spray production and secondary aerosol formation, at both high and low latitudes.

We believe this offers valuable insights that will assist in resolving the impacts of synoptic meteorology in models.

Further responses to the reviewer's comments have been included below. Please note that text coloured in red refers to the added text in the manuscript. All page and line numbers refer to the revised manuscript (CWT_Revised_Manuscript_20200501.docx) and supplementary material (CWT_Revised_Supplement_20200501.docx). If the text has been significantly changed, only the section number is given in this document (e.g. "Section 2.1").

**General comments**
*Referee's Comment*
*1. Page 4, line 5 - In my eyes stating that "the ship and its sampling facilities are discussed in detail" elsewhere is not particularly helpful. I would like to see at least the basic information presented here alongside the data.*

Author's Answer
1. The authors acknowledge the benefit of providing this detail in the manuscript and Section 2.1 has been expanded with details of the sampling system layout and conditions within the sampling lines. Details regarding carbon monoxide measurements were added to Section 2.3, and the following clarification was added on Page 4, line 10:

An integrated Nafion membrane drier maintained the was used to relative limit humidity at at < 40 % in the ACSM inlet the ACSM sampling inlet.

**Specific comments**
*Referee's Comment*
*1. Page 6, line 29 - "...number fraction towards these distributions" would read better as "number fraction towards these sizes".*

Author's Answer
1. Section 3.2 has now been extensively rewritten to reflect improvements to the mode fitting process and the above wording has been removed.

**Specific comments**
*Referee's Comment*
*Page 14 - Although this is perhaps more personal taste, in my view the conclusions section is rather more of a summary. The conclusions of the paper should be more concise than its current form.*

Author's Answer
1. The conclusion (Section 5) has been extensively rewritten. The result is more concise and avoids summarizing general observations in favour of restating the key findings

**Referee #2**

*The paper by Alroe et al. reports the result of a Southern Ocean cruise and marine aerosol features from the underexplored region. The paper is fluently written and was a pleasure to read, but unfortunately contains little scientific advancement beyond the current state of knowledge. The results broadly agree very well with the already published papers in the topical area, but I was wondering why the authors did not come up with unique insights possessing a good dataset and all the relevant instruments. Was the study not from the Southern Ocean and without thorough trajectory and ocean colour analysis I would have difficulty recommending it for publication, but I would like to encourage the authors to take a second look at their results and try to enhance the value of the paper by highlighting certain novel aspects. I hope my comments will motivate the authors and may help in improving the paper. For example, were there any oceanographic data available like in-situ chlorophyll measurements to infer phytoplankton blooms or more general tracer measurements by e.g. fluorometer? Last, but not least properly fitted size distributions may help to deepen the discussion of particle sources.*

Author's Answer

The authors appreciate the reviewer's comments that have helped us to enhance our investigation and refine the manuscript. On reflection, we agree that the scope of the original manuscript was broad and there were considerable opportunities for more detailed investigation. In response to both reviewers' comments, we have significantly increased the depth and quality of our analysis.

Changes to the analysis include:
- Improved aerosol size distribution mode fitting, including the separation of a nucleation mode throughout much of the voyage
- Modelling of sea salt mass concentrations using a wind speed-dependent model, to allow evaluation of the scaled sea salt measurements obtained from ACSM measurements, and to assess the validity of the model for the conditions observed during this voyage
- Assessing contributions to CCN and the impact of variations in SSA and Aitken mode aerosol concentrations
- Detailed analysis of back trajectories to identify meteorological conditions associated with the above variations in SSA and Aitken mode aerosol

Subsequent to the above changes, we found that the chlorophyll-a exposure over the rasterized productive region (discussed in the original manuscript) was insufficient to explain observed variations in Aitken and nucleation mode aerosol properties. Instead, we have extensively rewritten the manuscript with a greater focus on synoptic-scale meteorological conditions and their key influence on the variability of CCN concentrations in the Southern Ocean.

This has resulted in a much stronger study that highlights the following key findings:
1) Accurate modelling of sea spray aerosol concentrations must be based on detailed air mass meteorological histories, including parameters such as wind speed, rainfall frequency and intensity, and altitude above sea level relative to the marine boundary layer depth. In a dynamic environment like the Southern Ocean, these parameters must be considered along the trajectory of the air mass rather than at the sampling point, and sea spray aerosol are unlikely to reach equilibrium concentrations until the air mass has experienced favourable conditions (low rainfall and low altitudes) for approximately 48 hours.

2) Several periods were identified when CCN concentrations were significantly influenced by either increased wind-related SSA or large-diameter Aitken mode aerosol. Aitken mode aerosol were to be the most significant of these two sources of CCN, potentially providing up to 56% of CCN, even at high latitudes.
3) Despite the high wind speeds associated with frequent cyclonic systems at high latitudes, these weather systems ultimately limited SSA concentrations due to increased rates of precipitation, in-cloud scavenging and vertical transport of air masses into the free troposphere where they became decoupled from the surface source of sea spray. As a result, the highest SSA concentrations and SSA contributions to CCN were observed at lower latitudes.
4) Similarly, while there was frequent evidence of secondary sulfate formation, growth of these aerosol to CCN-relevant diameters was only observed where air masses transited over biologically productive regions at relatively low altitudes, remained within the MBL for 24-48 hours and then encountered a strong cold front which combined rainfall, compression of the MBL and vertical transport into the free troposphere.

Variability arising from subgrid scale meteorology and aerosol properties is poorly represented in current models (Weigum et al., 2016; Lin et al., 2017). This study now provides a detailed examination of the sources of variability in CCN during several episodes of long range transport, sea spray production and secondary aerosol formation, at both high and low latitudes. We believe this offers valuable insights that will assist in resolving the impacts of synoptic meteorology in models.

Further responses to the reviewer's comments have been included below. Please note that text coloured in red refers to the added text in the manuscript. All page and line numbers refer to the revised manuscript (CWT_Revised_Manuscript_20200501.docx) and supplementary material (CWT_Revised_Supplement_20200501.docx). If the text has been significantly changed, only the section number is given in this document (e.g. "Section 2.1").

**Key comments**
*Referee's Comment*
*1. Were there any oceanographic data available like in-situ chlorophyll measurements to infer phytoplankton blooms or more general tracer measurements by e.g. fluorometer?*

Author's Answer
The only in-situ oceanographic measurements included water temperature, salinity and conductivity. Since there were no water chemistry measurements, this study relies on chlorophyll estimates from MODIS-Aqua satellite observations of ocean colour. Usage of sea-surface dimethyl sulfide concentrations from the Surface Ocean Lower Atmosphere Study (SOLAS, Natural Environment Research Council, UK) dataset was also considered, but since this dataset represents a global climatology, it would not have assisted in localizing and assessing specific phytoplankton blooms relevant to this voyage.

*Referee's Comment*
*2. The authors used a sea salt tracer to estimate sea salt mass, but they could also crosscheck with SSA estimated mass and number from recent of earlier sea salt source functions at the observed wind speed. Something more can be done here if proper lab calibrations were not performed for ACSM which would have been very useful given instrumental differences between high resolution AMS used by Ovadnevaite et al. and ToF ACSM used in this study.*

Author's Answer

The authors agree that the ACSM measurements of sea salt required further support, given that the instrument was not calibrated for that species. Sea salt mass concentrations have now been estimated using the mass-based sea salt source function given in Ovadnevaite et al. (2012). Section 3.3 has rewritten with full details on how this model was implemented and a comparative evaluation of our observations. In short, it was found that the model offered reasonable agreement for favourable meteorological conditions where the air masses remained within the MBL for at least 48 hours prior to sampling, with rainfall rates below 0.25 mm h$^{-1}$. However, poor agreement was observed where the above conditions were not met. This suggested that our measurements of sea salt mass were reasonable and permitted increases in sea salt mass concentrations to be correlated against changes in CCN number concentrations. It also highlighted air mass meteorological history as an important consideration when estimating sea spray concentrations from wind speed-dependent models.

Other sea spray source functions were also investigated to model SSA number concentration, specifically those given in Blot et al. (2013) and Norris et al. (2013). Both models predicted unrealistically high SSA concentrations, frequently comparable to or higher than the observed $N_{10}$ concentrations. As discussed in Answer 4 below, an SSA mode could not be fitted to the size distributions, preventing any direct comparison of observed and modelled number concentrations. Therefore, we chose not to pursue this further and have focused our analysis on the mass-based model.

*Referee's Comment*

*3. It is important to realise that the observed particles and their size distributions are a product of cloud cycling. Or in other words, the observed size distribution is a product, not the source for cloud processing. Indeed, all accumulation mode particles can undergo subsequent cloud cycles, but Aitken mode cannot given their modal diameter of 30nm. Bi-modal size distribution has exactly arisen after cloud cycling. If there were no clouds accumulation mode chemical composition would have been entirely made of primary sea salt. As was noted at the beginning properly fitted size distributions (Comments regarding Figure 3) and the above comments may enhance the discussion.*

Author's Answer

3. We agree with these comments in general and have applied a more rigorous mode fitting process to separate the cloud-processed Aitken and accumulation modes from relatively unprocessed nucleation mode aerosol. Section 3.2 of the manuscript has been rewritten to reflect the improved mode fitting procedure and detail is also given in Author's Answer 4 below.

*Referee's Comment*

*4. The authors must redo the fitting of log-normal distributions. The shape of "bi-modal" distribution is clearly suggesting more modes, e.g. nucleation mode of ∼10nm and second accumulation mode centred at ∼420nm. I do not see symmetrical log-normal modes, only one side of them which makes me wondering what the inventive fitting was applied. Specifically, significant departure of log-normal mode from the observed size distribution is suggesting additional mode(s) together with the potentially excessive geometric spread (sigma) above ∼1.5. Consequently, nucleation mode is suggesting contribution from new particle formation at the coast or open ocean depending on the trajectories. But first, please do the proper fitting.*

Author's Answer

4. The authors agreed that the fitting process could be improved. The original fitting process used a similar process as described in Modini et al. (2015), in which multiple component lognormal modes with unconstrained means are fitted to the distribution. The resulting component modes are then grouped to form each of the expected particle size distribution modes (e.g nucleation, Aitken, accumulation and SSA). However, for this study, it was not possible to fit an SSA mode because the SMPS distributions were limited to maximum diameter of 662 nm. This did not offer enough diameter bins to adequately constrain the SSA mode and separate it from the tail of the larger accumulation mode. In addition, concentrations at the smallest diameters were often noisy. As a result of both limitations, the above unconstrained process often resulted in fitted modes with distribution parameters (mean, spread, amplitude) that did not correspond to physically meaningful modes.

To overcome this, a revised fitting process has been applied and is detailed in Section 3.2. In brief, a maximum of three modes were permitted. Their spread parameters were limited based on values reported in other studies (e.g. (Modini et al., 2015; Fossum et al., 2018) and their means were constrained to diameter ranges that best reflected the nucleation, Aitken and accumulation modes observed in the unfitted distributions (Fig. 2a). Broadening from the SSA mode often became apparent at diameters larger than 300 nm, so fits were only applied to the diameter range 7 – 300 nm to limit this bias. This yielded much improved modal fits (Figs. 4 and S3) and revealed a nucleation mode which was frequently seen in *mSO* air masses. In turn, these new fitted modes have permitted investigation of meteorological conditions and biologically productive regions that favoured strong populations of nucleation mode aerosol, or growth of large diameter Aitken mode aerosol, and their corresponding impact on CCN concentrations (Section 4.2.1, Figs. S5-7).

**Minor comments**

*Referee's Comment*

*1. Page 4. Line 8. I could not find in this paper nor the referenced paper information about the sampling inlet. Was it community sampling duct of certain dimensions with individual instruments sub-sampling from it? What was the total flow and laminar conditions?*
*Was it experiencing a significant drying during air passage through it?*

Author's Answer

1. The following details regarding the sampling inlet and sample flow conditions have been added to Section 2.1 (Page 3, line 23):

"Aerosol sampling was performed through a common sampling inlet mounted on a mast, located approximately 18 m above sea level at the bow of the ship and co-located with a suite of meteorological instruments. Sample air was drawn through this inlet at a flow rate of approximately 420 L min$^{-1}$. The sample passed through a 161.5 mm stainless steel tube to a manifold in the bow of the ship, 8 m below the mast in the ship's bow. Most aerosol instruments sampled from this manifold through 3/8" stainless steel tubing. The remaining flow was passed through a 32 mm stainless steel tube to a manifold in a second laboratory and the ACSM sampled from this secondary manifold through 1/4" stainless steel tubing."

There was an ambient temperature differential of up to 25 degrees between the laboratories and the outdoor conditions. The sample air likely underwent significant drying during transit

through the sampling ducts. However, the air temperature in the main sampling laboratory was not accurately controlled or monitored and sample humidity was only measured after the membrane dryers, so the drying effect associated with the temperature change cannot be evaluated separately.

*Referee's Comment*
*2. Page 4, line 24. Was the drier used for ACSM to limit the excessive humidity?*

Author's Answer
2. The ACSM is designed with an integrated membrane dryer that was used throughout the voyage. Section 2.2 has been updated with the following text (Page 4, line 10):
"An integrated Nafion membrane drier maintained the relative humidity at <40 % in the ACSM inlet."

*Referee's Comment*
*3. Page 6, line 13. BC threshold was set conservatively in general, however, given pristine nature of the Southern Ocean may have been set even lower, especially as it was an hourly average. Also while number concentration criterion is appropriate, number concentration is not a great arbiter when new particle formation events may increase N10 concentration substantially. It would very useful to present BC data in Figure 2.*

Author's Answer
3. The authors agree that high $N_{10}$ concentrations cannot be directly linked to contamination by ship exhaust emissions. After further evaluation, this criterion was found to be unnecessary and has been removed. Likewise, we agree that the Southern Ocean may be expected to have lower BC concentrations than the baseline level used in this study. The MAAP is certainly capable of detecting lower concentrations when using 1-hour averages. Assuming an inverse square relationship between the 95 % confidence interval detection limit and the averaging interval, the detection limit for 1-hour averaged measurements is estimated at 8 ng m$^{-3}$. However, the goal of this study was not to isolate the most pristine periods, but rather to characterize the conditions commonly present in the Southern Ocean, which include periods influenced by the long-range transport of continental aerosol from major land masses. Therefore the threshold of 30 ng m$^{-3}$ was selected on the basis of BC concentrations reported in other studies of Southern Ocean marine air masses, typically ranging between 1–70 ng m$^{-3}$ (e.g. (Schmale et al., 2019; Humphries et al., 2015; Weller et al., 2013; Kim et al., 2017; Cravigan et al., 2015). This range of values likely reflects differences in instrumentation and sampling location and the selected threshold of 30 ng m$^{-3}$ lies below the midpoint of this range. After applying this threshold, all remaining peaks in BC concentration were examined with respect to wind speed and direction relative to the ship's heading, the path of the associated back trajectories, CO and radon concentrations. In any cases where the ship and air mass trajectories indicated likely contamination, or where CO concentrations were elevated without direct correlation with increased radon concentrations, the associated data was excluded from further analysis. Section 3.1 has been re-written to reflect this reasoning and additional analysis. Also Section 2.3 has been modified to include details regarding the CO measurements (Page 4, line 23):

"Black carbon (BC) and carbon monoxide (CO) concentrations were measured with a Thermo Fisher Scientific 5012 Multi-Angle Absorption Photometer (MAAP) and an Aerodyne

Research Inc. infrared Laser Trace Gas Monitor, respectively. The trace gas monitor was not adequately calibrated and, on average, the CO measurements were 19 ± 5 ppb higher than the estimated reference concentrations given by the Global Greenhouse Gas Reference Network (GLOBALVIEW-CO, 2009). This indicates that the measurements were not quantitatively accurate but correlations between BC and CO were examined when identifying periods contaminated by ship emissions."

*Referee's Comment*
*4. Page 6, line 28. I am confused with this statement as sea spray usually contributes significant number fraction to submicron aerosol based on several established sea spray source functions. Possibly the authors were meant to say that the supermicron sea spray mode contributed a small number fraction to the observed size distributions which would be true.*

Author's Answer
4. Section 3.2 has now been extensively rewritten (as discussed in Key Comments Authors Answer 4, above) and this statement has been removed.

*Referee's Comment*
*5. Page 7, line 28. Please provide ranges of temperature and wind speed.*

Author's Answer
5. The following details have been added to Section 4.1 (Page 7, line 9):

Throughout the voyage, the median air and sea surface temperatures were 4.7 °C (IQR: 2.9-8.7 °C) and 3.9 °C (IQR: 2.6-8.9 °C), respectively. Median wind speeds observed at the ship were 12.0 m s$^{-1}$ (IQR: 9.5-13.9 ms$^{-1}$)

*Referee's Comment*
*6. Page 8, line 31. Notation is confusing with that of sulfur species. Perhaps is better using upper script notation or mSO-first, mSO-second and so on.*

Author's Answer
6. Thank you for this suggestion. The authors agree and this notation has been updated throughout the manuscript using Roman numerals (e.g. *mSO-IV*).

*Referee's Comment*
*7. Page 9, line 5. Why mSO4 is preceding mSO3 period and mSO3 preceding mSO2?*

Author's Answer
7. Section 4.2 has been extensively rewritten and now does not discuss each marine air mass as separate cases.

*Referee's Comment*
*8. Page 10, line 19. There is no mention nor reason why CCN not reported for mSO periods.*

Author's Answer
8. Section 4.2 has been extensively rewritten with a specific focus on CCN in the marine air masses.

*Referee's Comment*
*9. Page 11, line 29. The only Antarctic terrestrial sources of BC are the scientific bases.*

Author's Answer
9. The authors agree and on further investigation, it seems possible that BC was detected from both the Durmont D'Urville research station (6:00 AM on 9th February) and Mawson research station (early on 10th February), due to small peaks in BC concentration as air mass trajectories transitioned westward past these sites. However, these were very brief events and had negligible impact on other aerosol properties. Since the sources of BC in Antarctic air masses are not a major focus of this study, we have removed that comment from Section 4.4.

*Referee's Comment*
*10. Page 12, line 21. This conclusion must be supported by ammonium concentration and degree of neutralisation (DON). There is no mention nor reason of the absence of ammonium concentration. ACSM is perfectly capable of measuring ammonium ion.*

Author's Answer
10. While the ACSM is technically capable of measuring ammonium ($NH_4$), during this voyage, the instrument was configured with a three-stage vacuum system that is less efficient than the more common four-stage system. This lead to increased detection limits for all species. In addition, $NH_4$ is a relatively "noisy" species due to signal interference with other common species with similar masses, and the mass concentration of non-refractory aerosol in the remote marine environment is typically quite low. As a result, the $NH_4$ signal was above the detection limit for only 5% of this voyage so it was not feasible to compare the $NH_4$ concentrations or DON between the air masses.

*Referee's Comment*
*11. Figure 2. BC concentration would be extremely useful on a separate scale of radon graph.*

Author's Answer
11. BC concentration has now been added as a third panel in Fig. 2c.

*Referee's Comment*
*12. Table 1. Air mass notations should be spelt below the Table or in the caption. NH4 is an important species discriminating between different neutralisation degrees depending on continental impact. DON could be presented too. DL should be noted below Table or simply like <0.XX*

Author's Answer
12. Air mass abbreviations have been defined in the caption for Table 1. Organic aerosol mass and BC detection limits have been specified in the table, as suggested. As discussed in Author's

Answer 10, above, NH4 concentrations were almost constantly below the detection limit and therefore DON could not be calculated. These two parameters have not been presented.

**References**

Blot, R., Clarke, A. D., Freitag, S., Kapustin, V., Howell, S. G., Jensen, J. B., Shank, L. M., McNaughton, C. S., and Brekhovskikh, V.: Ultrafine sea spray aerosol over the southeastern Pacific: open-ocean contributions to marine boundary layer CCN, Atmos. Chem. Phys., 13, 7263-7278, doi: 10.5194/acp-13-7263-2013, 2013.

Cravigan, L. T., Ristovski, Z., Modini, R. L., Keywood, M. D., and Gras, J. L.: Observation of sea-salt fraction in sub-100 nm diameter particles at Cape Grim, J. Geophys. Res.-Atmos., 120, 1848-1864, doi: 10.1002/2014JD022601, 2015.

Fossum, K. N., Ovadnevaite, J., Ceburnis, D., Dall'Osto, M., Marullo, S., Bellacicco, M., Simó, R., Liu, D., Flynn, M., Zuend, A., and O'Dowd, C.: Summertime Primary and Secondary Contributions to Southern Ocean Cloud Condensation Nuclei, Sci. Rep.-UK, 8, 13844, doi: 10.1038/s41598-018-32047-4, 2018.

GLOBALVIEW-CO: Co-operative Atmospheric Data Integration Project - Carbon Monoxide, CD-ROM, NOAA ESRL, Boulder, Colorado, USA, [Also available on Internet via anonymous FTP to aftp.cmdl.noaa.gov, Path: products/globalview/co], 2009.

Humphries, R. S., Schofield, R., Keywood, M. D., Ward, J., Pierce, J. R., Gionfriddo, C. M., Tate, M. T., Krabbenhoft, D. P., Galbally, I. E., Molloy, S. B., Klekociuk, A. R., Johnston, P. V., Kreher, K., Thomas, A. J., Robinson, A. D., Harris, N. R. P., Johnson, R., and Wilson, S. R.: Boundary layer new particle formation over East Antarctic sea ice – possible Hg-driven nucleation?, Atmos. Chem. Phys., 15, 13339-13364, doi: 10.5194/acp-15-13339-2015, 2015.

Kim, J., Yoon, Y. J., Gim, Y., Kang, H. J., Choi, J. H., Park, K. T., and Lee, B. Y.: Seasonal variations in physical characteristics of aerosol particles at the King Sejong Station, Antarctic Peninsula, Atmos. Chem. Phys., 17, 12985-12999, doi: 10.5194/acp-17-12985-2017, 2017.

Lin, G., Qian, Y., Yan, H., Zhao, C., Ghan, S. J., Easter, R., and Zhang, K.: Quantification of marine aerosol subgrid variability and its correlation with clouds based on high-resolution regional modeling, J. Geophys. Res.-Atmos., 122, 6329-6346, doi: 10.1002/2017jd026567, 2017.

Modini, R. L., Frossard, A. A., Ahlm, L., Russell, L. M., Corrigan, C. E., Roberts, G. C., Hawkins, L. N., Schroder, J. C., Bertram, A. K., Zhao, R., Lee, A. K. Y., Abbatt, J. P. D., Lin, J., Nenes, A., Wang, Z., Wonaschütz, A., Sorooshian, A., Noone, K. J., Jonsson, H., Seinfeld, J. H., Toom-Sauntry, D., Macdonald, A. M., and Leaitch, W. R.: Primary marine aerosol-cloud interactions off the coast of California, J. Geophys. Res.-Atmos., 120, 4282-4303, doi: 10.1002/2014jd022963, 2015.

Norris, S. J., Brooks, I. M., Moat, B. I., Yelland, M. J., de Leeuw, G., Pascal, R. W., and Brooks, B.: Near-surface measurements of sea spray aerosol production over whitecaps in the open ocean, Ocean Sci., 9, 133, doi: http://dx.doi.org/10.5194/os-9-133-2013, 2013.

Ovadnevaite, J., Ceburnis, D., Canagaratna, M., Berresheim, H., Bialek, J., Martucci, G., Worsnop, D. R., and O'Dowd, C.: On the effect of wind speed on submicron sea salt mass concentrations and source fluxes, J. Geophys. Res.-Atmos., 117, doi: 10.1029/2011jd017379, 2012.

Schmale, J., Baccarini, A., Thurnherr, I., Henning, S., Efraim, A., Regayre, L., Bolas, C., Hartmann, M., Welti, A., Lehtipalo, K., Aemisegger, F., Tatzelt, C., Landwehr, S., Modini, R. L., Tummon, F., Johnson, J., Harris, N., Schnaiter, M., Toffoli, A., Derkani, M.,

Bukowiecki, N., Stratmann, F., Dommen, J., Baltensperger, U., Wernli, H., Rosenfeld, D., Gysel-Beer, M., and Carslaw, K.: Overview of the Antarctic Circumnavigation Expedition: Study of Preindustrial-like Aerosols and Their Climate Effects (ACE-SPACE), B. Am. Meteorol. Soc., doi: 10.1175/bams-d-18-0187.1, 2019.

Weigum, N., Schutgens, N., and Stier, P.: Effect of aerosol subgrid variability on aerosol optical depth and cloud condensation nuclei: implications for global aerosol modelling, Atmos. Chem. Phys., 16, 13619-13639, doi: 10.5194/acp-16-13619-2016, 2016.

Weller, R., Minikin, A., Petzold, A., Wagenbach, D., and König-Langlo, G.: Characterization of long-term and seasonal variations of black carbon (BC) concentrations at Neumayer, Antarctica, Atmos. Chem. Phys., 13, 1579-1590, doi: 10.5194/acp-13-1579-2013, 2013.

---

## Author Response (AR2)

**Referee #2**

*I was glad to see that the authors made a good use of suggestions and encouragement to do a more in-depth analysis of their study results and to draw perfectly matching conclusions with great insights for the reader. The paper improved very substantially and apart from few minor corrections can be accepted for publication.*

Authors' Answer

The authors thank the reviewer for the favourable feedback. We agree with their advice and have made the following changes to the manuscript. All page and line numbers refer to the revised manuscript (CWT_Revised_Manuscript_20200603.pdf). All text coloured in red represent additions to the manuscript, or section numbers corresponding to more extensive changes.

**Minor comments**

*Referee's Comment 1*

*Abstract, line 21. It is not the storms which inhibited sea spray (quite contrary, in fact), but rather precipitation and developing sea state. It is well explained in the text and especially conclusions, but short summary became somewhat misleading.*

Authors' Answer

We agree that the original wording was unclear. The sentence has been replaced as follows:

Abstract, Line 20:

Frequent cyclonic weather conditions were observed at high latitudes and the associated strong wind speeds led to predictions of high concentrations of sea spray aerosol. However, these modelled concentrations were not achieved due to increased aerosol scavenging rates from precipitation and convective transport into the free troposphere, which decoupled the air mass from the sea spray flux at the ocean surface.

*Referee's Comment 2*

*Page 5, lines 21-25. SSA is misrepresented here because of existence of several submicron SSA modes going down to even nanometer size sea spray. I believe this section applies to larger size SSA, but has to be noted accordingly.*

Authors' Answer

It is true that the SSA aerosol exist even at very small diameters. The SSA mode is usually fit on the basis of the aerosol size distribution at larger diameters (>500nm) to avoid bias from the non-SSA modes and is approximated with a very broad mode which attempts to capture the SSA contribution at small diameters. However, since the SMPS distributions only extended to 660nm, the SSA contribution could not be reliably separated from the other modes. Therefore, we acknowledge that some unknown SSA fraction may be present in the fitted accumulation, Aitken and nucleation modes. The first two paragraphs of Section 3.2 have been rewritten for clarity and to acknowledge these limitations.

*Referee's Comment 3*

*Page 6, line 37. That is suggesting under-developed sea state and possibly precipitation in air masses prior to reaching ship position. Again conclusions are more informative than the text itself.*

Authors' Answer

As indicated in the preceding paragraph, there was minimal rainfall along the 48-hour air mass trajectories for this period. However, we concur that the sea state may have been underdeveloped, as indicated by the improved agreement between the observed sea salt mass concentrations and the parameterisation for increasing wind speed. We have added this interpretation with the following concluding sentences at Page 7, Lines 3-5:

Furthermore, it suggests that wind speeds in this region were increasing or variable, leading to an under-developed sea state with lower SSA production rates than would be expected for constant or decreasing wind speeds. This highlights the impact of air mass meteorological history and synoptic-scale weather conditions on SSA concentrations.

*Referee's Comment 4*
*Page 10, line 20. This is typical katabatic wind regime.*

Authors' Answer
We agree that this observation is not unexpected and have re-expressed the sentence in a more concise manner (Page 10, Lines 22-24):

The air masses consistently followed the typical katabatic wind regime (Humphries et al., 2016), remaining within the free troposphere while over the continent and then rapidly subsiding into the MBL as they moved offshore over the coastal region (Fig. S10).

Additional Authors' Comments
We identified a miscalculation in the comparisons between modelled and observed sea salt mass concentrations, discussed in Section 3.3. At Page 6, Line 39 the revised calculations indicate that the steady-state parameterisation generated sea salt mass concentrations that exceeded the concentrations by 61% (rather than 49%). At Page 7, Line 1, concentrations from the increasing wind speed parameterisation exceeded the observations by 13% (rather than 4%). These values have been updated in the manuscript.

[revised manuscript text omitted]